# The value of satellite soil moisture and snow cover data for the transfer of hydrological model parameters to ungauged sites

Rui Tong[1,2], Juraj Parajka[1,2], Borbála Széles[1,2], Isabella Greimeister-Pfeil[1,3], Mariette Vreugdenhil[3], Jürgen Komma[2], Peter Valent[2,4] and Günter Blöschl[1,2]

[1]Centre for Water Resource Systems, TU Wien, Vienna 1040, Austria
[2]Institute of Hydraulic Engineering and Water Resources Management, TU Wien, Vienna 1040, Austria
[3]Department of Geodesy and Geoinformation, TU Wien, Vienna 1040, Austria
[4]Department of Land and Water Resources Management, Slovak University of Technology in Bratislava, Bratislava 810 05, Slovakia

*Correspondence to*: Rui Tong (tong@hydro.tuwien.ac.at)

**Abstract.**

The recent advances in remote sensing provide opportunities for more reliably estimating the parameters of conceptual hydrologic models. However, the question of whether and to what extent the use of satellite data in model calibration may assist in transferring model parameters to ungauged catchments has not been fully resolved. The aim of this study is to evaluate the efficiency of different methods for transferring model parameters obtained by multiple objective calibrations to ungauged sites and to assess the model performance in terms of runoff, soil moisture, and snow cover predictions relative to existing regionalization approaches. The model parameters are calibrated to daily runoff, satellite soil moisture (ASCAT), and snow cover (MODIS) data. The assessment is based on 213 catchments situated in different physiographic and climate zones of Austria. For the transfer of model parameters, eight methods (global and local variants of arithmetic mean, regression, spatial proximity, and similarity) are examined in two periods, i.e., the period in which the model is calibrated (2000-2010) and an independent validation period (2010-2014). The predictive accuracy is evaluated by leave-one-out cross-validation. The results show that the method by which the model is calibrated in the gauged catchment has a larger impact on runoff prediction accuracy in the ungauged catchments than the choice of the parameter transfer method. The best transfer methods are global and local similarity and the kriging approach. The performance of the transfer methods differs between lowland and alpine catchments. While the soil moisture and snow cover prediction efficiencies are higher in lowland catchments, the runoff prediction efficiency is higher in alpine catchments. A comparison of model transfer methods based on parameters calibrated to runoff, snow cover, and soil moisture with those based on parameters calibrated to runoff only indicates that the former outperforms the latter in terms of simulating soil moisture and snow cover. The performance of simulating runoff is similar, and the accuracy depends mainly on the weight given to the runoff objective in the multiple objective calibrations.

## 1 Introduction

Prediction of runoff hydrographs is one of the main scientific and practical hydrological applications. Understanding and representation of the processes and their interactions during runoff generation and their sensitivity to the controls are one of the unsolved problems in hydrology (Blöschl et al., 2019). It is also essential for many purposes of societal relevance, such as engineering design, for water resources management, hydropower operation or for risk assessment (Sachs and McArthur, 2005; Blöschl and Montanari, 2010; Kovacs et al., 2012; Parajka et al., 2013). However, in most places of interest, no runoff data are directly observed, so the runoff prediction needs to be based on additional information observed in the region or transferred from other gauged catchments.

Numerous studies have explored and evaluated methods for prediction of runoff in ungauged sites (He et al., 2011; Hrachowitz et al., 2013; Blöschl et al., 2013). The most frequently used method consists of applying a hydrological model driven by model parameters derived in catchments with runoff observations. The review and synthesis of studies and applications of hydrological models in ungauged sites presented in Parajka et al. (2013) and recently Guo et al. (2021) indicate that there is a variety of transfer approaches tested in different parts of the world, covering different climates, altitudes or landscape settings. These studies' main outcome is that the runoff predictions in ungauged catchments tend to be more accurate in humid than in arid regions and more accurate in large than in small catchments (Parajka et al, 2013). However, the selection and performance of methods differ between studies, and there is no general recommendation for the choice of the approach. There seems to be a consensus that the spatial proximity and similarity methods perform better in humid regions (Guo et al., 2021). In arid catchments similarity and parameter regression methods tend to be applied more frequently and perform slightly better (Parajka et al., 2013; Yang et al. 2018, 2020).

Recent studies have explored the role and impact of gauge density on the efficiency of the methods (Parajka et al., 2015; Lebecherel et al., 2016; Neri et al. 2020), as well as the advanced definition of similarity measures based on spatial patterns (Li and Zhang, 2017; Beck et al., 2020; Narbondo et al., 2020), or catchment response characteristics (Tegegne and Kim, 2018). Along with these investigations, one of the recent focuses in hydrological modelling evaluates the use of observations additional to runoff (Bouaziz et al., 2021). Multiple objective calibrations can help constrain hydrological models, reduce uncertainty, and improve hydrological predictions (Efstratiadis and Koutsoyiannis, 2010; Rakovec et al., 2016; Dembélé et al., 2020). Most of the previous studies investigated the potential of calibrating hydrological model parameters by using different runoff signatures or using some additional hydrological characteristics such as snow cover, cover, soil moisture, evaporation, groundwater level or their combinations (please see the review in Tong et al., 2021). However, only a few studies have investigated the application of multiple objective approaches for evaluating the transfer of model parameters to ungauged sites. Parajka et al. (2005 and 2006), examined the value of snow depth observation and scatterometer soil moisture measurements for improving hydrological simulations in ungauged catchments and showed that the use of scatterometer resulted in more consistent patterns of soil moisture estimates, but it did not improve the runoff model efficiencies in ungauged catchments. Zhang et al. (2020) recently used remotely sensed evapotranspiration to calibrate the hydrological model parameters without

streamflow observations and found the streamflow-free calibration could be satisfactory for monthly and mean annual runoff simulation in the humid catchments. Huang et al. (2020) verified the effectiveness of this approach for the regionalization with spatial proximity method in ungagged basins. They reported that using the bias-corrected remotely sensed evapotranspiration has great potential in estimating daily and monthly runoff. However, the role and impact of using additional data for the prediction of daily hydrographs in ungauged sites are still not well understood. Such assessment is essential in particular for projecting impacts of changing climate or land use conditions on the hydrological cycle in general and specifically on runoff generation. The aim of this study is to investigate the value of using model parameters obtained by multiple objective calibrations for daily hydrological predictions in ungauged sites. Specifically, we test the efficiency of different methods for transfer of such model parameters to ungauged sites and evaluate the model performance to observations of runoff and remotely sensed estimates of soil moisture and snow cover. We extend the results of Tong et al. (2021) who assessed the value of Metop ASCAT and MODIS satellites for calibration of conceptual hydrological models and to test to what extent it improves the performance of existing regionalization approaches. The analysis and comparison of model simulations of runoff, soil moisture and snow cover are performed for a large sample of catchments situated in different physiographic and climate zones of Austria, so allows evaluating the potential of the application of remotely sensed data at the regional scale. The effect of multi-objective calibration with the use of satellite soil moisture and snow cover data on model parameter transferability are hence assessed.

## 2 Method

### 2.1 Hydrological model

The transfer of the model parameters is evaluated for a conceptual hydrological model (TUWmodel, Viglione and Parajka, 2020). The TUWmodel is a variant of the HBV model (Bergström 1992; Parajka et al., 2007) implemented in the R-environment (Astagneau et al., 2020). This study uses a semi-distributed version in which the inputs and outputs of the model are processed for elevation zones of 200 meters, but model parameters are assumed lumped in each catchment. Such setting allows to effectively account for spatial variability of rainfall and melt processes, particularly in alpine regions, and to keep the number of parameters small which simplifies their transfer to ungauged sites. For comparison with soil moisture satellite data, we used a single soil (root-zone) layer version of TUWmodel. It simulates changes in snow, root zone and groundwater storages in each elevation zone. The model runs on a daily time step and combines three routines: snow routine, soil moisture routine, and river flow routing routine. The snow routine uses a degree-day concept to reflect snow accumulation and melt with a degree-day factor and a threshold melt temperature. The snowfall part of precipitation and snow accumulation is calculated by using snow and rain threshold temperatures. The soil moisture routine represents changes in the soil moisture state of the root zone due to evapotranspiration and runoff generation.

$$S_{SM,i} = S_{SM,i-1} + \left( PR + M - E_A \right) \cdot \varDelta t \qquad (1)$$

where $S_{SM}$ is the root zone soil moisture, which controls runoff generation and actual evaporation $E_A$, $PR$ is rain, $M$ is snowmelt, and $\Delta t$ is the time step (one day). The contribution $\Delta S_{UZ}$ of rain and snowmelt to runoff is calculated by an explicit scheme as a function of the $S_{SM}$ using a non-linear relationship controlled by two model parameters, maximum soil storage $FC$ and nonlinearity parameter $BETA$ controlling characteristics of runoff generation

$$\Delta S_{UZ} = (\theta)^{BETA}(PR + M) \cdot \Delta t \qquad (2)$$

$$\theta = \frac{S_{SM}}{FC} \qquad (3)$$

Actual evapotranspiration is estimated from potential evaporation (model input) and a model parameter representing the soil moisture state above which the actual equals potential evaporation. For a comparison with satellite soil moisture estimates, simulated root zone soil moisture is scaled by the maximum soil moisture storage $FC$ model parameter and represents hence the relative root zone soil moisture $\theta$.

The runoff routing module represents routing on the hillslopes and river flow routing in the stream. The runoff response function consists of two reservoirs, representing the upper and lower storage zones. The outflows from reservoirs in each elevation zone is summed up and routed by a triangular transfer function.

The model involves 15 model parameters. They are automatically calibrated using the multi-objective calibration strategy of Tong et al., (2021). The joint objective function $O_F$ consists of weighting three individual parts related to runoff ($O_Q$), soil

moisture ($O_{SM}$), and snow cover ($O_{SC}$).

$$O_F = w_Q \cdot O_Q + w_{SM} \cdot O_{SM} + w_{SC} \cdot O_{SC} \qquad (4)$$

where $w_Q$ is the weight to the runoff objective function, $w_{SM}$ and $w_{SC}$ are the weights to the soil moisture and snow cover objectives, respectively. Multiple objective approaches for each regionalization method are examined for 11 runoff weights $w_Q$ (ranging from 0.0 to 1.0 with a step of 0.1). The combinations of weights are taken from Tong et al. (2021), where the

115 weights of soil moisture and snow are equal, and the total sum of all weights is 1.0. More details about the combinations of the objective functions are presented in Tong et al. (2021).

The runoff objective function $O_Q$ emphasizes both high and low flows (Parajka and Blöschl, 2008) and is described by a compound objective function combing two variants of Nash-Sutcliffe coefficients, i.e. it is estimated from observed and logarithmic-transformed river flow values (Nash and Sutcliffe, 1970):

$$O_Q = 0.5 \cdot NSE + 0.5 \cdot NSE_{log} \qquad (5)$$

$$NSE = 1 - \frac{\sum_{i=1}^{n}(Q_{obs,i} - Q_{sim,i})^2}{\sum_{i=1}^{n}(Q_{obs,i} - \overline{Q_{obs}})^2} \qquad (6)$$

$$NSE_{\log} = 1 - \frac{\sum\limits_{i=1}^{n}\left(\log(Q_{obs,i}) - \log(Q_{sim,i})\right)^2}{\sum\limits_{i=1}^{n}\left(\log(Q_{obs,i}) - \log(\overline{Q_{obs}})\right)^2} \qquad (7)$$

where $Q_{obs,i}$ and $Q_{sim,i}$ are the river flow observation and simulation of day $i$.

The measure of soil moisture agreement ($O_{SM}$) is determined by the Pearson correlation coefficient ($O_{SM}$) between the satellite soil water index (SWI) which estimates the saturation degree of the root zone (Wagner et al., 1999b; Silvestro et al., 2015) and simulated relative soil moisture in each elevation zone:

$$O_{SM} = \frac{\sum\limits_{i=1}^{N_{days}} \sum\limits_{j=1}^{N_{zones}} \left((\theta_{sim,i,j} - \overline{\theta_{sim}})(\theta_{obs,i,j} - \overline{\theta_{obs}})\right)}{\sqrt{\sum\limits_{i=1}^{N_{days}} \sum\limits_{j=1}^{N_{zones}} \left((\theta_{sim,i,j} - \overline{\theta_{sim}})^2 (\theta_{obs,i,j} - \overline{\theta_{obs}})^2\right)}} \qquad (8)$$

where $\theta_{sim,i,j}$ is the simulated relative soil moisture from the hydrological model and $\theta_{obs,i,j}$ is the averaged value of the observed Soil Water Index (SWI) from ASCAT pixels for the day $i$ and elevation zone $j$. $\overline{\theta_{sim}}$ and $\overline{\theta_{obs}}$ are the mean value of the simulations and observations for the days and elevation zones which are not masked in the ASCAT SWI product due to presence of snow or frozen ground. The rationale behind selecting the Pearson correlation as a measure of agreement is that it assesses the spatial and temporal correspondence of the satellite soil moisture (assumed as ground truth) and simulated root zone soil moisture time series. At the spatial resolution of original ASCAT dataset (ca. 12.5 km), the satellite estimates of root zone soil moisture reflect more the regional rainfall and melt processes patterns, and are thus more closely related to altitudinal zonality than to morphometric characteristics of terrain that operate at smaller scales. The calculation of $O_{SM}$ from soil moisture averages for elevation zones thus allows representing the agreement in regional and seasonal soil moisture patterns. Choice of a correlation coefficient has the advantage of not being sensitive to the units. In a preliminary analysis, we tested different ways for calculation of the $O_{SM}$ combining soil moisture estimated from different elevation zones better describes the agreement than the correlation between soil moisture estimates averaged at the catchments scale (see Supplement, Fig. S3). Particularly in the alpine regions, correlation calculated from the catchment averages masks the spatial variability in the agreement between ASCAT and hydrologic root zone soil moisture estimates. A similar approach has also been used in previous studies (e.g., Parajka et al., 2006; Gruber et al., 2020; Beck et al., 2021). A more detailed description of the calculation of soil moisture agreement is presented in the Supplement.

The snow cover objective function $O_{SC}$ minimises the sum of snow overestimation $S_O$ and underestimation $S_U$ errors (Parajka and Blöschl, 2008):

$$O_{SC} = 1 - (S_O + S_U) \qquad (9)$$

The snow overestimation error shows the percentage of the condition if the snow is simulated from the model, but the snow cover is not retrieved by the satellite (MODIS):

$$150 \quad S_O = \frac{1}{\sum\limits_{i=1}^{N_{days}} \sum\limits_{j=1}^{N_{zones}} A_{i,j}} \sum\limits_{i=1}^{N_{days}} \sum\limits_{j=1}^{N_{zones}} A_{i,j} \bigcap (SWE_{i,j} > \xi_{SWE}) \bigcap (SCA_{i,j} = 0) \qquad (10)$$

where $A_{i,j}$ is the area of zone $j$ that is cloud-free according to MODIS observation on the day $i$. $SWE_{i,j}$ is the simulated snow water equivalent in elevation zone $j$ greater than a threshold ($\xi_{SWE}$) 10mm, $SCA_{i,j}$ is the MODIS snow covered area within this zone, and $N_{days}$ is the number of days with cloud cover less than a threshold ($\xi_C$) 50%.

The snow underestimation error indicates the percentage of the condition if no snow is simulated, but snow cover retrieved by the MODIS is over a threshold ($\xi_{SCA}$) of 25% in the zone, i.e.:

$$S_U = \frac{1}{\sum\limits_{i=1}^{N_{days}} \sum\limits_{j=1}^{N_{zones}} A_{i,j}} \sum\limits_{i=1}^{N_{days}} \sum\limits_{j=1}^{N_{zones}} A_{i,j} \bigcap (SWE_{i,j} = 0) \bigcap (SCA_{i,j} > \xi_{SCA}) \qquad (11)$$

The thresholds of $\xi_{SWE}$, $\xi_C$, and $\xi_{SCA}$ were determined by the sensitivity analysis of Parajka and Blöschl (2008).

## 2.2 Transfer of model parameters to ungauged sites

The hydrological predictions at the ungauged sites are, in this study, based on model simulations driven by model parameters transferred from the gauged locations where the model has been calibrated, i.e. the sites with runoff observations. The model performances in terms of runoff, soil moisture, and snow with the calibrated parameters in the catchments assumed to be gauged can refer to Tong et al. (2021). For the transfer (i.e. regionalization), four groups of methods are evaluated (Table 1). The first group estimates model parameters as the arithmetic mean of all calibrated values in the study region (termed "global mean") or, alternatively, as the arithmetic mean of model parameters within a radius of 50 km from the catchment of interest (termed "local mean"). This arithmetic mean regionalization approach assumes similarity of all catchments within a specified radius where differences in the parameter values are caused only by random factors.

In the second group, the model parameters for ungauged sites are independently estimated from linear regressions between calibrated model parameters and catchment attributes. Similarly, as in the first group, two approaches are tested. The global multiple linear regression uses attributes and model parameters from all gauged catchments. The local multiple linear regression is applied within a 50 km search radius from an ungauged site. In all cases, the regression coefficients are estimated by the ordinary least squares method. For consistency with previous studies, a set of three catchment attributes associated with the largest multiple correlation coefficient for each ungauged site and each model parameter is used. To avoid multicollinearity, the variance inflation factor (Hirsch et al., 1992) is examined. For the transfer of model parameters, such a regression model is used, which has the largest correlation coefficient for the inflation factor less than 10.

The third group of transfer methods is based on the spatial proximity (or spatial distance) between the ungauged and the gauged catchments. The spatial distance between the two catchments is characterized by the distance between the respective catchment centroids. We test two methods of this group: the inverse distance weighting and the ordinary kriging. In both methods, individual parameters from several donor catchments are independently interpolated to a centroid of the ungauged catchment and then combined and used in the hydrological model. The power parameter in inverse distance interpolation is set to two.

The ordinary kriging method is based on a fixed exponential variogram with a nugget of 10% of the observed variance, a sill equal to the variance, and a range of 60 km. The test calculations in previous studies (Merz and Blöschl, 2004; Parajka et al., 2005) showed that this setting is consistent with the empirical variograms of most of the calibrated model parameters.

The fourth group of methods is based on the similarity between catchments with runoff observations and ungauged sites. The main idea of the similarity group of methods is to find for an ungauged site a donor catchment that is most similar in terms of
185 certain catchment attributes. The entire collection of model parameters calibrated for a donor catchment is then transferred to the ungauged site. The similarity is defined by a similarity index Φ (Burn and Boorman, 1993; Merz and Blöschl, 2004),

$$\Phi = \sum_{i=1}^{k} \frac{\left| x_i^G - x_i^U \right|}{\Delta X_i} \qquad (12)$$

where $X^G$ represents a vector of the normalized catchment attributes of the gauged (donor) catchments; $X^U$ are the normalized attributes of the ungauged catchment; and $\Delta X$ is the normalized range of attributes. In previous studies (e.g. Parajka et al.,
2005) a large number of catchment attributes and their combination have been tested in the study region. Based on the results and preliminary analyses (not shown here) for this study, we selected the approach with the best results. This variant is based on an *a priori* defined combination of the following catchment attributes: mean catchment elevation, stream network density, lake attenuation index and areal proportion of porous aquifers, land use, soils and geologic units. Similar to other approaches, also in this approach, we examine two variants. While the global similarity combination uses all study catchments for
estimation of the similarity, the local similarity combination estimates the similarity only within 50 km radius around the ungauged site.

**Table 1 The list of model parameter transfer methods tested in the study.**

| Group | Transfer method | Abbreviation |
|---|---|---|
| Arithmetic mean | Global mean | GM |
| | Local mean | LM |
| Regression | Global regression | GR |
| | Local regression | LR |
| Spatial proximity | Inverse distance weighting | ID |
| | Kriging | KR |
| Similarity | Global combination of physiographic attributes | GS |
| | Local combination of physiographic attributes | LS |

### 2.3 Evaluation of the prediction accuracy

The performance and efficiency of parameter transfer methods are evaluated by leave-one-out cross validation. Each catchment with observed runoff is considered in turn as ungauged and the transfer methods are used to estimate the parameter sets from other gauged catchments. The hydrological model is applied to simulate daily runoff, soil moisture and snow cover in the ungauged catchment. These simulations are then compared with the observations. The accuracy is quantified by three objectives $O_Q$, $O_{SM}$, and $O_{SC}$ in two periods, i.e. in the period used for model calibration (2000-2010, 2007-2010 for $O_{SM}$) and in an independent validation period (2010-2014). The efficiencies of the transferred model parameters are estimated for eleven different calibration variants (i.e. weight given to runoff) and compared to the efficiency obtained by a transfer of model parameters calibrated to runoff only.

### 3 Data

### 3.1 Study region

The study region is Austria which represents a wide range of physiographic conditions. The topography varies from flat land in the East and North to Alpine terrain in the West and South. Mean annual precipitation is less than 400 mm/year in the East and more than 2500 mm/year in the West. Land use is mainly agricultural in the lowlands and forest in the medium elevation ranges. Alpine vegetation and rocks prevail in the highest alpine regions.

The analysis is carried out for 213 catchments (Fig. 1). These catchments have been selected following previous studies (Viglione et al., 2013, Sleziak et al., 2020, Tong et al., 2021) and represent catchments with no significant anthropogenic effects on the water balance. The size of the catchments ranges from 13.7 to 6214 km² and the averaged slope varies from 1.74% to 43.91%. As previous studies have shown that the performance of regionalization methods differs between climatic zones (Parajka et al., 2013; Yang et al., 2020), separation of the effect of elevation and climate on the results was deemed important and the catchments were split into two groups representing drier lowland and hilly regions (catchments with mean elevation below 900 m a.s.l.) and wetter alpine conditions (catchments with mean elevation above 900 m a.s.l.). Out of the 213 catchments, 94 are classified as lowland catchments, and 119 as alpine catchments (Fig. 1). The climatic statistics of the two groups are presented in Table 2. The threshold of 900 m is chosen as a compromise between balancing the number of catchments in the groups and representing different physiographic regions.

**[Figure 1]**

**Table 2** Statistics of the climatic attributes of the 94 lowland catchments and 119 alpine catchments. With abbreviation, unit, minimum, maximum, and median. The standard deviations refer to spatial variability within each catchment.

| Attribute | Abbrev. | Unit | Lowland (mean elevation under 900 m a.s.l.) | | | Alpine (mean elevation over 900 m a.s.l.) | | |
|---|---|---|---|---|---|---|---|---|
| | | | Min. | Max. | Median | Min. | Max. | Median |
| Mean annual precipitation | MAP | mm | 728.13 | 1828.40 | 999.46 | 913.66 | 2301.84 | 1476.64 |
| Standard deviation of annual MAP | SDAP | mm | 10.79 | 367.57 | 71.49 | 30.13 | 289.87 | 152.90 |
| Mean air temperature | MAT | °C | 7.26 | 10.30 | 8.98 | -2.83 | 8.07 | 5.76 |
| Standard deviation of MAT | SDAT | °C | 0.06 | 1.71 | 0.57 | 0.40 | 3.55 | 1.64 |
| Mean annual potential evaporation | MEPI | mm | 618.36 | 740.45 | 690.08 | 233.49 | 657.01 | 563.00 |
| Standard deviation of MEPI | SDEPI | mm | 4.33 | 77.41 | 25.25 | 21.70 | 162.07 | 83.33 |
| Catchment aridity index (MEPI/MAP) | CAI | - | 0.36 | 0.98 | 0.66 | 0.18 | 0.69 | 0.37 |
| Standard deviation of aridity index | SDAI | - | 0.01 | 0.31 | 0.06 | 0.02 | 0.18 | 0.09 |

## 3.2 Hydrologic and climate data

The runoff data have been obtained from Central Hydrographical Bureau (HZB, ehyd.gv.at). The analysis period is from Sep. 2000 to Aug. 2014, which is split into the calibration (September 2000-August 2010) and validation periods (September 2010-August 2014).

Model inputs (i.e. mean of daily climate characteristics for elevation zones) are derived from the SPARTACUS gridded dataset (Hiebl and Frei, 2016, 2018). This dataset includes grid maps of maximum and minimum daily air temperature and precipitation with a spatial resolution of 1km. Daily mean air temperature is estimated as the mean between minimum and maximum air temperature. Potential evaporation model input is estimated by the Blaney-Criddle approach (Parajka et al., 2003). This approach estimates potential evaporation from mean daily air temperature and a potential sunshine duration index, which is calculated from a 1 km digital elevation model of Austria.

## 3.3 MODIS snow cover

The snow cover maps used in model calibration and regionalization validation are based on the combination of the daily, 500m resolution, Terra (MOD10A1) and Aqua (MYD10A1) MODIS datasets (Hall and Riggs, 2016a, 2016b). We use the latest collection 6 snow cover products, which includes the Normalized Difference Snow Index (NDSI). Snow cover mapping from MODIS products is performed in two steps. In the first step, NDSI pixels are classified into snow and land cover classes based on seasonally varying NDSI thresholds (Tong et al., 2020). In the second step, resulting the snow cover maps from Aqua and Terra products are combined to reduce the impact of clouds (Parajka and Blöschl, 2008). Finally, for each elevation zone of each catchment, the frequency of pixels classified as clouds, snow and land is calculated. This allows estimating the percent snow cover area of each catchment needed for the calculation of the snow cover model objective function.

## 3.4 ASCAT soil moisture

The satellite soil moisture data used in this study is the Soil Water Index (SWI) derived from an experimental version of the upcoming Disaggregated Metop ASCAT Surface Soil Moisture v2 product (H28) provided by the EUMETSAT Satellite Application Facility on Support to Operational Hydrology and Water Management (H SAF). The original ASCAT surface soil moisture dataset at 12.5 km (before disaggregation) is based on a new parameterization for the vegetation correction (Hahn et al., 2020), which has shown improved performance over Austria (Pfeil et al., 2018). The disaggregation process consists of a directional resampling method utilizing the connection between regional (12.5 km) and local (0.5 km) scale Sentinel-1 backscatter observations describing temporally stable soil moisture patterns also reflected in the radar backscatter measurements (Wagner et al. 2008). Surface and root zone soil moisture are available, where the root zone soil moisture is represented by the Soil Water Index (SWI), which is determined by an exponential filter introduced by Wagner et al. (1999a,b) and Albergel et al. (2008) with a characteristic time lag (T). The T-value represents the smoothing of soil moisture dynamics by infiltration, with higher T-values corresponding to a higher degree of smoothing. In order not to lose information on short-term soil moisture dynamics still present in deeper soil layers, T should be carefully chosen. Paulik et al (2014) compared the ASCAT SWI dataset to in situ soil moisture and found that the SWI agrees better with in situ soil moisture from deeper layers than the original surface soil moisture dataset. Moreover, they related the T-value with soil depth layers and found that the T-values 10 and 20 led to the highest correlations in the shallow subsurface (around 0-20 cm). To prevent the loss of short-term soil moisture dynamics, T-value=10 days was selected in this study. Besides, to exclude invalid ASCAT measurements affected by snow and frozen ground, soil moisture is masked as no data when soil temperatures at a soil depth of 0-7 cm are below 1°C or snow cover exceeds 30 % of the pixel with the information from the ECMWF Copernicus Climate Service (C3S) ERA5-Land.

## 4 Results

### 4.1 Efficiency of transfer methods to simulate runoff

Figure 2 and 3 show the median and scatter (i.e. 25% and 75% quantile) of runoff leave-one-out cross-validation efficiency for eight parameter transfer methods (panels) and 11 calibration variants (i.e. different runoff weight $w_Q$ used in model calibration) in the calibration (Fig. 2) and validation (Fig. 3)periods. The red symbols indicate the at site runoff efficiency estimated in Tong et al. (2021) in the calibration and validation periods. Panels on the left and right show the results for the lowland and alpine group of catchments, respectively. The results for the runoff weight $w_Q$ =1.0 represent the case when the model is calibrated to runoff only. The case $w_Q$=0.0 represents the case when the model is calibrated to satellite soil moisture and snow cover without using observed runoff.

The results show that the differences between the transfer methods are smaller than those between the different calibration variants, i.e. the different methods of calibrating the model in gauged catchments, for weights below 0.4. The impact of the

choice of calibration variant (weight on runoff $w_Q$) is smaller if the $w_Q$ is larger than 0.4. For $w_Q$ larger than 0.4 the differences between the transfer methods are larger, the differences between the transfer methods are larger, and the choice of transfer method is more important than that of the calibration variant. The worst parameter transfer (i.e. regionalization) methods are the global mean and the local regression approach. The median of runoff efficiency is particularly low, i.e. between 0.24 and 0.41, for calibration variants using $w_Q$ <0.3 in the calibration period. If $w_Q$ is larger than 0.7, the median of runoff efficiency of global mean and local regression is between 0.42 and 0.5 for the lowland and between 0.61-0.63 for the alpine catchments. The best transfer methods are global and local similarity and kriging interpolation. If the $w_Q$ is larger than 0.4, the median efficiency is between 0.67 and 0.69 in the lowland and between 0.71 and 0.74 in the alpine region. The efficiency of the transfer of model parameters calibrated by multiple objective approaches (for $w_Q$>0.4) for the similarity and kriging methods is the same as that for the transfer of model parameters obtained by calibration to runoff only ($w_Q$=1). The reason for the similar runoff efficiency of the regionalization methods between $w_Q$=0.4 to 1.0 is that the runoff model efficiencies of the donor catchments do not change obviously when the satellite soil moisture and snow cover were both included in the calibration with this $w_Q$ range (Tong et al., 2021). In the validation period (Fig. 3), the median of multiple objective calibrations ($w_Q$=0.8) of kriging in the lowland and similarity in the alpine catchments is even larger than the runoff efficiency obtained by a transfer (kriging or similarity) based on parameters calibrated to runoff only (variant $w_Q$=1). Additionally, Table 3 also shows that the regional variability (runoff model efficiency between catchments) is small for kriging while large for local regression methods. A comparison of local and global variants of the transfer methods indicates that the local methods are only slightly better than the global methods in terms of runoff efficiency. The largest difference between the local and global methods occurs for the mean approach, but the runoff efficiency of the local mean is noticeably lower than for the spatial proximity or similarity approaches. An exception is the regression of model parameters, which has a larger runoff efficiency for the global than the local approach. The reason is a larger correlation between model parameters and catchment attributes estimated from all catchments. For example, for $w_Q$=0.4, the median of the correlation between model parameters and catchment attributes for the local regression varies between 0.22 and 0.65. For the global regression approach, the median is larger and varies between 0.70 and 0.88. The results also show that the performance of transfer methods is better in the alpine than in the lowland catchments. In both groups, however, similarity and kriging are the best approaches for predicting daily runoff.

[Figure 2]

[Figure 3]

**Table 3. Scatter (difference of 75% and 25% quantiles of model efficiency) of leave-one-out runoff model efficiency (Eq. 5) obtained by eight groups of parameter transfer methods and eleven calibration weights for lowland (94) and alpine (119) catchments in the calibration (2000-2010, first value) and validation (2010-2014, second value) periods.**

| $w_Q$ | | 0.0 | 0.1 | 0.2 | 0.3 | 0.4 | 0.5 | 0.6 | 0.7 | 0.8 | 0.9 | 1.0 |
|---|---|---|---|---|---|---|---|---|---|---|---|---|
| Lowlands | GM | 0.17/0.32 | 0.22/0.56 | 0.25/0.47 | 0.23/0.39 | 0.22/0.34 | 0.21/0.34 | 0.22/0.35 | 0.21/0.32 | 0.21/0.31 | 0.20/0.30 | 0.21/0.31 |
| | LM | 0.32/0.29 | 0.15/0.27 | 0.15/0.22 | 0.14/0.24 | 0.15/0.23 | 0.15/0.19 | 0.19/0.21 | 0.17/0.22 | 0.16/0.25 | 0.17/0.25 | 0.18/0.27 |
| | GR | 0.26/0.28 | 0.27/0.66 | 0.18/0.37 | 0.25/0.37 | 0.22/0.33 | 0.20/0.31 | 0.24/0.38 | 0.18/0.24 | 0.17/0.21 | 0.15/0.26 | 0.17/0.32 |
| | LR | 0.94/1.82 | 0.51/0.85 | 0.88/1.45 | 0.56/0.72 | 0.56/0.55 | 1.10/1.61 | 0.63/1.05 | 0.68/1.15 | 0.58/0.86 | 0.61/0.84 | 0.61/0.84 |
| | ID | 0.23/0.30 | 0.13/0.29 | 0.13/0.25 | 0.14/0.22 | 0.14/0.20 | 0.15/0.23 | 0.15/0.23 | 0.15/0.21 | 0.15/0.23 | 0.15/0.22 | 0.16/0.23 |
| | KR | 0.45/0.47 | 0.13/0.19 | 0.12/0.16 | 0.13/0.16 | 0.13/0.15 | 0.15/0.19 | 0.14/0.18 | 0.12/0.17 | 0.12/0.18 | 0.13/0.18 | 0.13/0.21 |
| | GS | 0.50/0.57 | 0.12/0.15 | 0.12/0.13 | 0.14/0.15 | 0.15/0.12 | 0.15/0.13 | 0.16/0.12 | 0.17/0.11 | 0.17/0.11 | 0.17/0.11 | 0.17/0.13 |
| | LS | 0.53/0.57 | 0.12/0.15 | 0.12/0.14 | 0.14/0.16 | 0.15/0.14 | 0.15/0.13 | 0.16/0.12 | 0.16/0.11 | 0.17/0.12 | 0.17/0.11 | 0.17/0.12 |
| Alpine | GM | 0.22/0.29 | 0.21/0.26 | 0.18/0.24 | 0.20/0.26 | 0.18/0.23 | 0.19/0.24 | 0.18/0.25 | 0.19/0.24 | 0.19/0.23 | 0.19/0.22 | 0.18/0.22 |
| | LM | 0.29/0.35 | 0.23/0.24 | 0.19/0.19 | 0.18/0.19 | 0.14/0.17 | 0.15/0.18 | 0.15/0.18 | 0.15/0.19 | 0.15/0.20 | 0.15/0.19 | 0.15/0.19 |
| | GR | 0.29/0.35 | 0.26/0.25 | 0.20/0.24 | 0.18/0.24 | 0.19/0.21 | 0.18/0.18 | 0.19/0.17 | 0.19/0.17 | 0.20/0.19 | 0.16/0.16 | 0.17/0.18 |
| | LR | 0.96/0.99 | 0.51/0.59 | 0.61/0.61 | 0.58/0.61 | 0.45/0.46 | 0.35/0.37 | 0.38/0.51 | 0.37/0.43 | 0.57/0.71 | 0.38/0.45 | 0.48/0.53 |
| | ID | 0.24/0.34 | 0.25/0.25 | 0.21/0.21 | 0.19/0.19 | 0.16/0.19 | 0.16/0.17 | 0.16/0.18 | 0.15/0.17 | 0.17/0.17 | 0.16/0.18 | 0.15/0.17 |
| | KR | 0.33/0.46 | 0.21/0.23 | 0.18/0.20 | 0.15/0.18 | 0.13/0.16 | 0.12/0.16 | 0.11/0.15 | 0.11/0.17 | 0.11/0.15 | 0.11/0.15 | 0.11/0.14 |
| | GS | 0.48/0.59 | 0.24/0.22 | 0.21/0.22 | 0.17/0.20 | 0.14/0.17 | 0.16/0.19 | 0.17/0.21 | 0.16/0.22 | 0.17/0.20 | 0.17/0.21 | 0.18/0.23 |
| | LS | 0.50/0.53 | 0.23/0.20 | 0.22/0.22 | 0.17/0.18 | 0.13/0.16 | 0.16/0.17 | 0.15/0.20 | 0.16/0.18 | 0.16/0.19 | 0.14/0.19 | 0.16/0.21 |

## 4.2 Efficiency of transfer methods to simulate soil moisture

The evaluation of eight parameter transfer methods to simulate root zone soil moisture is presented in Fig. 4 and 5. The results show the median and scatter (i.e. 25% and 75% quantile) of correlation between the satellite root zone soil moisture index and simulated relative root zone soil moisture in lowland (left two panels) and alpine (right two panels) catchments. The comparison of different transfer methods and calibration variants indicates that the difference between the transfer methods is similar as that found for the prediction of daily runoff. The best transfer methods are kriging and similarity (local and global) approaches. In the lowland catchments, the median of soil moisture correlation ranges between 0.62 and 0.70 for these methods. The impact of the calibration variants is, for each transfer method, smaller than found for runoff. Generally, the correlation increases with decreasing $w_Q$ and the soil moisture agreement tends to be larger if the $w_Q$ is smaller than 0.4. A much smaller agreement (correlation) between soil moisture estimates is found in the alpine catchments, this may likely be due to the heterogeneity in temperature and snow cover in mountainous regions when the soil moisture is retrieved from the satellite (Tong et al., 2021). The best transfer methods in alpine catchments are local and global similarity and kriging. The median correlation between modelled and satellite soil moisture is however small and varies between 0.14 and 0.22. From Table 4, the regional variability of soil moisture correlation is similar for each method and larger in the lowlands than that in the alpine regions. The comparison between the correlations of the calibration and validation periods shows a similar pattern. Interestingly, in the alpine catchments, the validation period correlations are slightly higher than those found for the transfer

in the calibration period. This is likely related to the warmer validation period. Warming decreases snow cover area particularly in the alpine regions and hence decreases the frequency of pixels which need to be masked in the soil moisture dataset.

**[Figure 4]**

**[Figure 5]**

**Table 4. Scatter (difference of 75% and 25% quantiles of model efficiency) of leave-one-out soil moisture correlation (Eq. 8) obtained by eight groups of parameter transfer methods and eleven calibration weights for lowland (94) and alpine (119) catchments in the calibration (2007-2010, first value) and validation (2010-2014, second value) periods.**

| | $w_Q$ | 0.0 | 0.1 | 0.2 | 0.3 | 0.4 | 0.5 | 0.6 | 0.7 | 0.8 | 0.9 | 1.0 |
|---|---|---|---|---|---|---|---|---|---|---|---|---|
| Lowlands | GM | 0.24/0.22 | 0.24/0.22 | 0.25/0.22 | 0.25/0.23 | 0.26/0.24 | 0.26/0.23 | 0.27/0.24 | 0.28/0.24 | 0.28/0.24 | 0.29/0.24 | 0.28/0.24 |
| | LM | 0.30/0.31 | 0.31/0.28 | 0.32/0.28 | 0.32/0.29 | 0.32/0.29 | 0.32/0.28 | 0.32/0.28 | 0.31/0.26 | 0.31/0.26 | 0.31/0.25 | 0.30/0.25 |
| | GR | 0.29/0.24 | 0.23/0.26 | 0.29/0.26 | 0.29/0.31 | 0.27/0.27 | 0.30/0.23 | 0.30/0.27 | 0.30/0.25 | 0.30/0.25 | 0.30/0.24 | 0.28/0.24 |
| | LR | 0.28/0.26 | 0.27/0.24 | 0.27/0.22 | 0.30/0.25 | 0.31/0.29 | 0.34/0.34 | 0.33/0.31 | 0.27/0.25 | 0.27/0.25 | 0.30/0.29 | 0.30/0.29 |
| | ID | 0.29/0.27 | 0.28/0.27 | 0.28/0.27 | 0.29/0.28 | 0.30/0.28 | 0.30/0.28 | 0.29/0.28 | 0.29/0.25 | 0.31/0.24 | 0.30/0.24 | 0.29/0.24 |
| | KR | 0.29/0.26 | 0.29/0.26 | 0.29/0.27 | 0.29/0.26 | 0.29/0.27 | 0.29/0.28 | 0.29/0.29 | 0.28/0.25 | 0.31/0.25 | 0.29/0.25 | 0.30/0.26 |
| | GS | 0.29/0.26 | 0.30/0.28 | 0.30/0.27 | 0.30/0.26 | 0.30/0.26 | 0.30/0.27 | 0.31/0.28 | 0.30/0.27 | 0.30/0.26 | 0.30/0.24 | 0.27/0.25 |
| | LS | 0.29/0.26 | 0.31/0.28 | 0.31/0.27 | 0.30/0.27 | 0.31/0.26 | 0.30/0.27 | 0.31/0.28 | 0.31/0.27 | 0.33/0.27 | 0.31/0.26 | 0.31/0.27 |
| Alpine | GM | 0.23/0.19 | 0.23/0.18 | 0.23/0.19 | 0.24/0.19 | 0.24/0.19 | 0.23/0.20 | 0.23/0.20 | 0.23/0.19 | 0.23/0.19 | 0.23/0.19 | 0.22/0.19 |
| | LM | 0.26/0.25 | 0.20/0.25 | 0.18/0.25 | 0.17/0.26 | 0.17/0.26 | 0.21/0.26 | 0.19/0.26 | 0.20/0.26 | 0.21/0.24 | 0.21/0.24 | 0.22/0.23 |
| | GR | 0.22/0.21 | 0.16/0.25 | 0.14/0.39 | 0.15/0.37 | 0.17/0.26 | 0.18/0.28 | 0.20/0.24 | 0.16/0.25 | 0.20/0.28 | 0.20/0.22 | 0.19/0.24 |
| | LR | 0.27/0.32 | 0.22/0.30 | 0.17/0.37 | 0.19/0.37 | 0.16/0.34 | 0.20/0.30 | 0.22/0.30 | 0.21/0.39 | 0.16/0.37 | 0.16/0.35 | 0.15/0.34 |
| | ID | 0.25/0.24 | 0.24/0.25 | 0.21/0.26 | 0.20/0.25 | 0.17/0.24 | 0.21/0.27 | 0.23/0.28 | 0.22/0.28 | 0.21/0.25 | 0.20/0.23 | 0.21/0.22 |
| | KR | 0.23/0.29 | 0.20/0.29 | 0.21/0.28 | 0.21/0.29 | 0.19/0.27 | 0.21/0.30 | 0.22/0.28 | 0.19/0.28 | 0.21/0.26 | 0.20/0.25 | 0.21/0.23 |
| | GS | 0.24/0.29 | 0.20/0.28 | 0.23/0.25 | 0.18/0.25 | 0.16/0.24 | 0.17/0.23 | 0.18/0.23 | 0.18/0.24 | 0.18/0.21 | 0.17/0.20 | 0.17/0.19 |
| | LS | 0.22/0.31 | 0.20/0.28 | 0.24/0.26 | 0.20/0.25 | 0.18/0.24 | 0.19/0.25 | 0.20/0.24 | 0.20/0.24 | 0.20/0.21 | 0.19/0.21 | 0.16/0.19 |

### 4.3 Efficiency of transfer methods to simulate snow cover

The efficiency of eight transfer methods to simulate snow cover is evaluated in Fig. 6 and 7. The results indicate that the variability and differences between the regionalization approaches are the smallest for snow efficiency. A much larger difference and impact on snow efficiency has the runoff weight used in model calibration. The difference in snow efficiency between transfer methods for the same $w_Q$ is mostly between 1 and 3%, but the snow efficiency for different $w_Q$ (and the same transfer method) ranges between 8 and 17%. The snow efficiency decreases with increasing $w_Q$, and it is generally larger in

the lowland than in the alpine catchments. It is related to the different frequencies of snow cover conditions in these two regions, generally the snow free condition is with less error for the simulation. Interestingly, in the alpine catchments, the similarity-based approaches and kriging have the smallest efficiency, and the most accurate results are obtained by the global mean and global regression methods. At the same time, the regional variability of the snow model efficiencies (Table 5) has a small difference between different $w_Q$. Also indicated in the table 5, the local regression method performed relatively unstable

for simulating snow, and in the Alpine regions the regional variability of snow model efficiency is larger than that in the lowlands. The impact of the calibration variant is, however, much more important than the selection of transfer method. The comparison between calibration and validation periods indicates an overall larger snow efficiency in the validation period. This is likely linked with a warmer validation period and generally fewer days with snow cover associated with an increase in air temperature in the last decades (Duethmann and Blöschl, 2018).

[Figure 6]

[Figure 7]

Table 5. Scatter (difference of 75% and 25% quantiles of model efficiency) of leave-one-out snow model efficiency (Eq. 9) obtained by eight groups of parameter transfer methods and eleven calibration weights for lowland (94) and alpine (119) catchments in the calibration (2000-2010, first value) and validation (2010-2014, second value) periods.

| | $w_Q$ | 0.0 | 0.1 | 0.2 | 0.3 | 0.4 | 0.5 | 0.6 | 0.7 | 0.8 | 0.9 | 1.0 |
|---|---|---|---|---|---|---|---|---|---|---|---|---|
| Lowlands | GM | 0.05/0.02 | 0.05/0.03 | 0.06/0.04 | 0.06/0.05 | 0.06/0.05 | 0.06/0.06 | 0.06/0.06 | 0.06/0.06 | 0.06/0.06 | 0.06/0.06 | 0.06/0.06 |
| | LM | 0.04/0.03 | 0.04/0.02 | 0.06/0.04 | 0.07/0.05 | 0.08/0.07 | 0.08/0.07 | 0.08/0.07 | 0.08/0.07 | 0.08/0.06 | 0.07/0.07 | 0.06/0.07 |
| | GR | 0.05/0.03 | 0.05/0.02 | 0.06/0.04 | 0.07/0.06 | 0.06/0.06 | 0.06/0.06 | 0.07/0.07 | 0.06/0.06 | 0.07/0.06 | 0.07/0.06 | 0.07/0.07 |
| | LR | 0.04/0.03 | 0.05/0.03 | 0.07/0.05 | 0.10/0.07 | 0.13/0.12 | 0.10/0.09 | 0.09/0.07 | 0.08/0.09 | 0.10/0.09 | 0.09/0.08 | 0.09/0.08 |
| | ID | 0.04/0.03 | 0.04/0.02 | 0.06/0.04 | 0.06/0.05 | 0.06/0.05 | 0.06/0.05 | 0.06/0.05 | 0.07/0.06 | 0.07/0.06 | 0.07/0.06 | 0.07/0.06 |
| | KR | 0.04/0.02 | 0.04/0.02 | 0.06/0.04 | 0.06/0.05 | 0.06/0.05 | 0.06/0.05 | 0.06/0.05 | 0.07/0.05 | 0.07/0.05 | 0.07/0.05 | 0.07/0.06 |
| | GS | 0.04/0.03 | 0.04/0.03 | 0.07/0.04 | 0.07/0.05 | 0.06/0.05 | 0.07/0.05 | 0.07/0.05 | 0.06/0.05 | 0.07/0.05 | 0.07/0.05 | 0.07/0.06 |
| | LS | 0.04/0.03 | 0.04/0.03 | 0.07/0.04 | 0.07/0.05 | 0.06/0.05 | 0.06/0.05 | 0.07/0.05 | 0.07/0.06 | 0.07/0.06 | 0.08/0.06 | 0.07/0.06 |
| Alpine | GM | 0.09/0.08 | 0.09/0.08 | 0.10/0.09 | 0.12/0.10 | 0.12/0.11 | 0.13/0.11 | 0.13/0.12 | 0.13/0.12 | 0.13/0.12 | 0.13/0.12 | 0.14/0.12 |
| | LM | 0.10/0.10 | 0.12/0.11 | 0.13/0.12 | 0.15/0.12 | 0.15/0.13 | 0.16/0.14 | 0.17/0.14 | 0.17/0.14 | 0.17/0.14 | 0.16/0.14 | 0.16/0.14 |
| | GR | 0.11/0.11 | 0.11/0.11 | 0.14/0.12 | 0.14/0.12 | 0.15/0.12 | 0.15/0.13 | 0.16/0.15 | 0.17/0.15 | 0.16/0.15 | 0.16/0.14 | 0.15/0.14 |
| | LR | 0.14/0.12 | 0.17/0.14 | 0.20/0.19 | 0.17/0.14 | 0.18/0.17 | 0.18/0.18 | 0.19/0.18 | 0.20/0.18 | 0.16/0.16 | 0.17/0.15 | 0.18/0.17 |
| | ID | 0.10/0.11 | 0.13/0.12 | 0.15/0.14 | 0.16/0.13 | 0.16/0.14 | 0.16/0.14 | 0.16/0.14 | 0.16/0.14 | 0.16/0.14 | 0.16/0.14 | 0.16/0.14 |
| | KR | 0.11/0.11 | 0.14/0.14 | 0.17/0.15 | 0.17/0.14 | 0.16/0.15 | 0.16/0.16 | 0.16/0.15 | 0.15/0.14 | 0.16/0.15 | 0.17/0.15 | 0.16/0.14 |
| | GS | 0.11/0.10 | 0.15/0.14 | 0.16/0.16 | 0.18/0.15 | 0.17/0.14 | 0.16/0.14 | 0.16/0.14 | 0.17/0.14 | 0.14/0.13 | 0.16/0.13 | 0.16/0.14 |
| | LS | 0.11/0.10 | 0.14/0.13 | 0.16/0.15 | 0.18/0.16 | 0.18/0.14 | 0.17/0.14 | 0.17/0.15 | 0.17/0.14 | 0.16/0.13 | 0.16/0.13 | 0.15/0.13 |

**4.4 Improvement of multi objective regionalization vs runoff only regionalization**

This section analyses the impact of calibration and runoff weight on the efficiency of different transfer methods to simulate runoff, soil moisture and snow cover. We compared the efficiencies of the predictions obtained by transferring model parameters from multiple-objective calibration (i.e. $w_Q$ <1) with those obtained by parameters calibrated to runoff only ($w_Q$=1). The results are shown in Fig. 8 and Fig. 9 for the calibration and validation periods, respectively, in terms of the relative improvement in snow cover (x-axis), soil moisture (y-axis) and runoff (colour of symbol) efficiencies for eight transfer methods (panels) and ten calibration weights (symbol size) in the lowland (left) and alpine (right) catchments. Positive efficiencies indicate an improvement when using a multiple-objective calibration compared to a runoff-only calibration. The figures suggest that the runoff predictions are very similar (i.e. within a 1% range) if the runoff weight is larger than 0.5. The largest impact of the multiple objective calibration is found for the soil moisture prediction, where the improvement is larger than for snow cover. In the lowland catchments, the soil moisture improvement is about 1 to 7% for $w_Q$ larger than 0.4. In the case of snow cover, the improvement is within 5% when $w_Q$ larger than 0.4. The patterns of improvement for the best transfer methods (kriging and similarity approaches) are very similar. The improvement increases with decreasing runoff weight but at the expense of decreasing runoff efficiency. The largest improvement of soil moisture (i.e. over 25%) is found for the similarity approaches in the alpine catchments, but the overall soil moisture efficiency is rather low. The patterns of improvement are very consistent with those obtained for the transfer of model parameters in the validation period (Fig 9). The improvement in soil moisture and snow cover increases with decreasing of runoff weight and the improvement is larger in the alpine than the lowland catchments. The patterns of improvement are very consistent with those obtained for the transfer of model parameters in the validation period. The most noticeable finding is that the improvement in soil moisture and snow cover increases with decreasing runoff weight and the improvement is larger in the alpine than in the lowland catchments.

[Figure 8]

[Figure 9]

**5 Discussion**

The main aim of the study was to test to what extent satellite data can improve the prediction of daily runoff in ungauged catchments. Tong et al. (2021) showed that ASCAT and MODIS satellites in hydrological model calibration improve simulations of a conceptual hydrological model, and that the improvements of runoff and soil moisture simulations were larger in low elevation and agricultural catchments. Here, we tested different methods for transferring model parameters from gauged to ungauged sites in the study region (Merz and Blöschl, 2004, Parajka et al., 2005). We examined which method and to what

extent transferring model parameters calibrated to different objectives improves the prediction of runoff, soil moisture and snow cover at ungauged sites. The results showed that the improvement is large in the simulation of soil moisture and snow cover without a significant impact on runoff prediction accuracy. The assessment of the efficiencies between different transfer methods indicates that the similarity approach and kriging of model parameters are the best in the study region. This finding is in line with that of Yang et al. (2020) who concluded that the spatial proximity and similarity approaches are relatively better than the parameter average or regression method in Norwegian catchments. This finding is also entirely consistent with the results of Parajka et al. (2005), who calibrated the model parameters in a larger number of catchments but using only runoff and interpolated snow depth (but not soil moisture). The results and efficiency of other similarity combinations tested in Parajka et al. (2005) are slightly lower or very similar in terms of runoff, soil moisture and snow cover efficiency. The efficiency of global and local regression and arithmetic mean methods are very similar, even though here we use 107 catchments (33%) fewer than tested in Parajka et al. (2005). Overall, the lower prediction accuracy of global mean or local regression is also consistent with the global assessment of the accuracy of the transfer methods presented in Parajka et al. (2013).

The results of Tong et al. (2021) show that using only satellite soil moisture and snow cover data is not sufficient for calibrating a conceptual hydrological model in ungauged catchments. The lower runoff accuracy of the calibration variants with no or only small weight to runoff is also reflected in the performance of the transfer methods. Satellite soil moisture and snow cover data are very useful for constraining model parameters related to the simulation of snow cover and soil moisture (Nijzink et al., 2018, Tong et al., 2021). However, the use of runoff is still essential for the accurate prediction of the complete runoff hydrographs, this is consistent with recent studies which used remotely sensed evaporation and/or total water storage anomalies for daily timescale hydrograph simulation (Zhang et al., 2020; Dembélé et al., 2020; Hulsman et al., 2021; Zhang et al., 2021). This finding agrees with the previous assessment of using soil moisture estimates from ERS scatterometer observations to improve hydrological simulations in gauged and ungauged catchments (Parajka et al., 2006). The scatterometer data assimilation did not improve the prediction in ungauged sites but provided more consistent patterns of soil moisture estimates. The use of the experimental disaggregated ASCAT dataset showed that the detailed spatial and temporal resolution of satellite soil moisture improves the application and agreement of soil moisture estimates in smaller lowland catchments. Similarly, as found in Széles et al. (2020) for a small Austrian catchment, the use of soil moisture data has a larger impact on the overall consistency of the model simulations compared to snow cover observations.

The transfer of model parameters to ungauged sites and the efficiency of different approaches for predicting runoff hydrographs are affected by different sources of uncertainty. During the conceptualization of the analysis we considered the following potential sources of uncertainty: (a) model inputs; (b) model structure; (c) accuracy of satellite data; (d) model calibration and (e) model parameter regionalization. We considered the impact of sources (a) to (c) to be smaller than (d) and (e) for the following reasons. The uncertainty of model inputs (a) is generally mainly due to the spatial interpolation of point (precipitation and air temperature) observations, as catchment averages are needed for water balance reasons, and this is a topic that has traditionally attracted a lot of interest in hydrology (e.g. Faurès et al. 1995). In this study, the model inputs (mean daily precipitation and air temperature) are estimated from the gridded SPARTACUS dataset with a grid resolution of 1 km that is

small relative to the median catchment size 167 km². Hiebl and Frei (2018) show the accuracy of the precipitation interpolation used in SPARTACUS to be high, and the monthly biases to be very small (values are within ±2%). The cross-validation of the air temperature interpolation (Hiebl and Frei, 2016) indicates no systematic overestimation or underestimation, i.e. the compound mean error is 0 °C, the root mean square error is 1.4 °C.

Model structure (b) is of course more difficult to evaluate and previous studies in the context of regionalization performance (e.g. Petheram et al. (2012), Parajka et al., 2013, Yang et al., 2020) have shown that the simpler models are not superior to complex models (nor much worse) in predicting daily hydrographs in ungauged catchments and more generally the difference between hydrological models tends to be small (Petheram et al., 2012). Parajka et al. (2013) grouped models according to the number of model parameters and showed that the median of the regionalisation performance (Nash-Sutcliffe efficiency) for each group of models is around 0.65. Yang et al. (2020) compared four daily rainfall-runoff models (GR4J, WASMOD, HBV and XAJ, with 6, 8, 13, and 17 parameters) and reported that the difference in model structure has a smaller impact on the regionalization model performance than the difference in climate conditions. Yang et al. (2020) shows that the average Nash-Sutcliffe runoff efficiency values are, for the best regionalization method (Physical similarity methods with output averaging), larger than 0.6 for all tested model structures.

The evaluation of MODIS snow cover (source (c)) of Tong et al. (2021) indicates an overall classification accuracy of the most recent MODIS snow cover product of larger than 97% which implies much smaller uncertainties than most of the other sources. The accuracy assessment of the experimental S1ASCAT dataset at the regional scale is still work in progress. A preliminary assessment (Panic et al., 2020, https://presentations.copernicus.org/EGU2020/EGU2020-16222_presentation.pdf) demonstrates S1ASCAT to compare well with point-scale and area-representative in situ root zone measurements. The correlation between observed in situ (i.e. TDR soil network and Cosmic-Ray Neutron Probe) and S1ASCAT soil moisture is 0.59 and 0.51, respectively. These correlations are higher than those obtained between in situ and existing COPERNICUS soil moisture products (SSM 1km and SWI 1km), and it is to be expected that only a part of the differences between the data types is due to the satellite data, as also TDR soil probes and Cosmic-Ray Neutron Probes have some level of uncertainty.

We thus decided to focus on the uncertainties resulting from model calibration (source (d) and selection of regionalization method (source (e)). The impact of using different time periods for the prediction of runoff hydrographs is evaluated by the split-sample uncertainty assessment proposed by Klemes (1985). Regionalization studies typically refer only to regionalisation model efficiencies obtained for the same period as used for model calibration. Our results indicate that regionalization efficiencies obtained in an independent validation period generally show a small decrease (loss) in runoff model performance. The median of the loss in Nash-Sutcliffe efficiency varies between 0.02 and 0.07, depending on the regionalization method and calibration weight. In the lowlands the average median loss is 0.06 while it is 0.03 in the alpine basins. The results also show that the median loss of runoff efficiency tends to be smaller for multiple-objective variants (average median loss of 0.05) than for variants using parameters calibrated to runoff only (average median loss of 0.06). These results are consistent with Yang et al. (2020), who reported a small degradation of regionalization runoff performance from the calibration to the validation period. The largest relative improvement of soil moisture efficiency is found in alpine catchments (more than 70%),

but the absolute value of the correlations (on average 0.31) are still lower than in lowland catchments (average correlation 0.59). These numbers suggest that the differences in performance (which are an indicator of the uncertainties to be expected) are quite significant for uncertainty source (d).

The evaluation of the uncertainty of runoff prediction using different regionalization methods (source (e)) shows that the variability in medians of runoff regionalization efficiency is smaller between regionalization methods than between different calibration variants (i.e. runoff weights). For example, the standard deviation of the medians obtained for eleven runoff weights for the local similarity regionalization method in alpine catchments is 0.17. The standard deviation of the medians between eight regionalization methods ranges (depending on the runoff weight) between 0.04 and 0.11. The differences are somewhat

smaller in lowland catchments (i.e. the standard deviation of medians between runoff weights and regionalization methods are 0.14 and about 0.09, respectively).

## 6. Conclusions and outlook

This study shows that the recent advances in remote sensing of water balance components contribute to improving the hydrological predictions in ungauged catchments. The main improvements are in estimating soil moisture and snow cover

dynamics, mostly in alpine catchments. Future analyses may focus on assessing the value of satellite data for other types of regionalization approaches, such as regional calibration (Parajka et al., 2007) or multi-scale parameter regionalization methods (Samaniego et al. 2010, Kumar et al. 2013). It will also be interesting to evaluate how much runoff information is needed in addition to existing satellite products to improve and constrain the model predictions in ungauged basins. Such investigation can also include an analysis of the role of nested catchments in parameter transfer and the impact of stream gauge density on

the regionalization model performance.

## Data availability

The streamflow observation data can be obtained from HZB (https://ehyd.gv.at/ ,BMLRT, 2021). The meteorological data from the ZAMG are currently not open to the public, requests should be sent to klima@zamg.ac.at. The ASCAT soil moisture

data is available via Copernicus Global Land Service (https://land.copernicus.eu/). MODIS C6 snow cover products are from NASA National Snow & Ice Data Center (https://nsidc.org/). Processed ASCAT SWI and MODIS snow cover maps used in this study are available upon request. Landuse information is from Copernicus Land Monitoring Service (https://land.copernicus.eu/). The R package of TUWmodel can be downloaded from CRAN (https://cran.r-project.org/web/packages/TUWmodel/index.html).

## Author contribution

RT and JP conceived and designed the study, wrote the codes, performed the analyses, and prepared the manuscript. BS, JK and PV were responsible for the data management, including quality control, processing, and validating. IP and MV were

responsible for developing, processing and validation of the ASCAT soil moisture data. GB supervised the study and contributed to the study design and interpretation of the results. All authors took part in the discussion of the results and revisions of the paper.

**Competing interests**

The authors declare that they have no conflict of interest.

**Acknowledgment**

The authors would like to acknowledge financial support provided by the Austrian Science Funds (FWF) as part of the Vienna Doctoral Program on Water Resource Systems (DK W1219-N28), the Austrian Research Promotion Agency (FFG) through the BMon project (Contract No. 866031), and the VEGA grant agency under the contract No. VEGA 1/0632/19. They would like to thank Sebastian Hahn for his support in developing and processing the ASCAT soil moisture data. Rui Tong is grateful for the scholarship from China Scholarship Council (CSC).

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

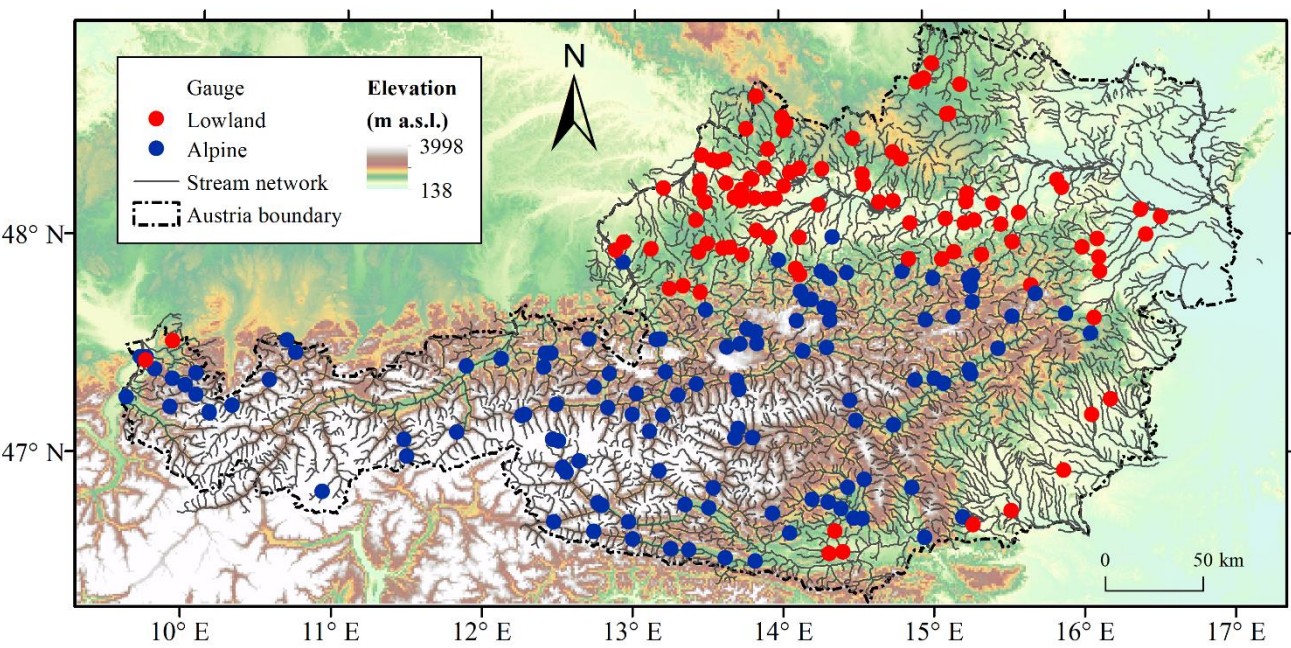

**Figure 1: Topography of the study region and location of 94 lowland (red symbols) and 119 alpine (blue symbols) catchments.**

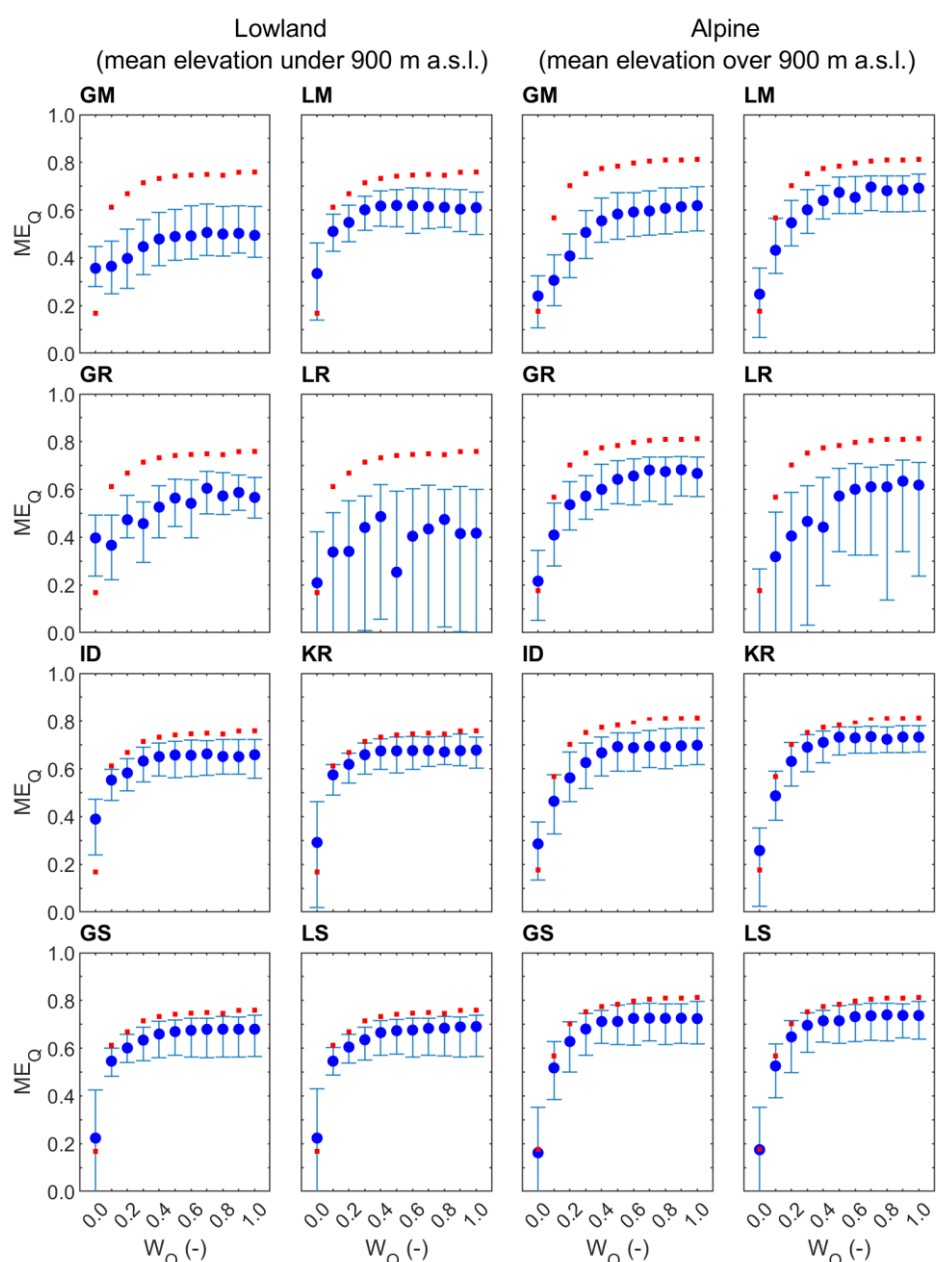

**Figure 2: Median and regional scatter of leave-one-out runoff model efficiency (Eq. 5) obtained by eight groups of parameter transfer methods and eleven calibration weights for lowland (94) and alpine (119) catchments in the calibration (2000-2010, blue symbols) period. Blue circles represent the median, whiskers the 25- and 75% quantiles of leave-one-out efficiency. Red symbols show the median of at site runoff model efficiency.**

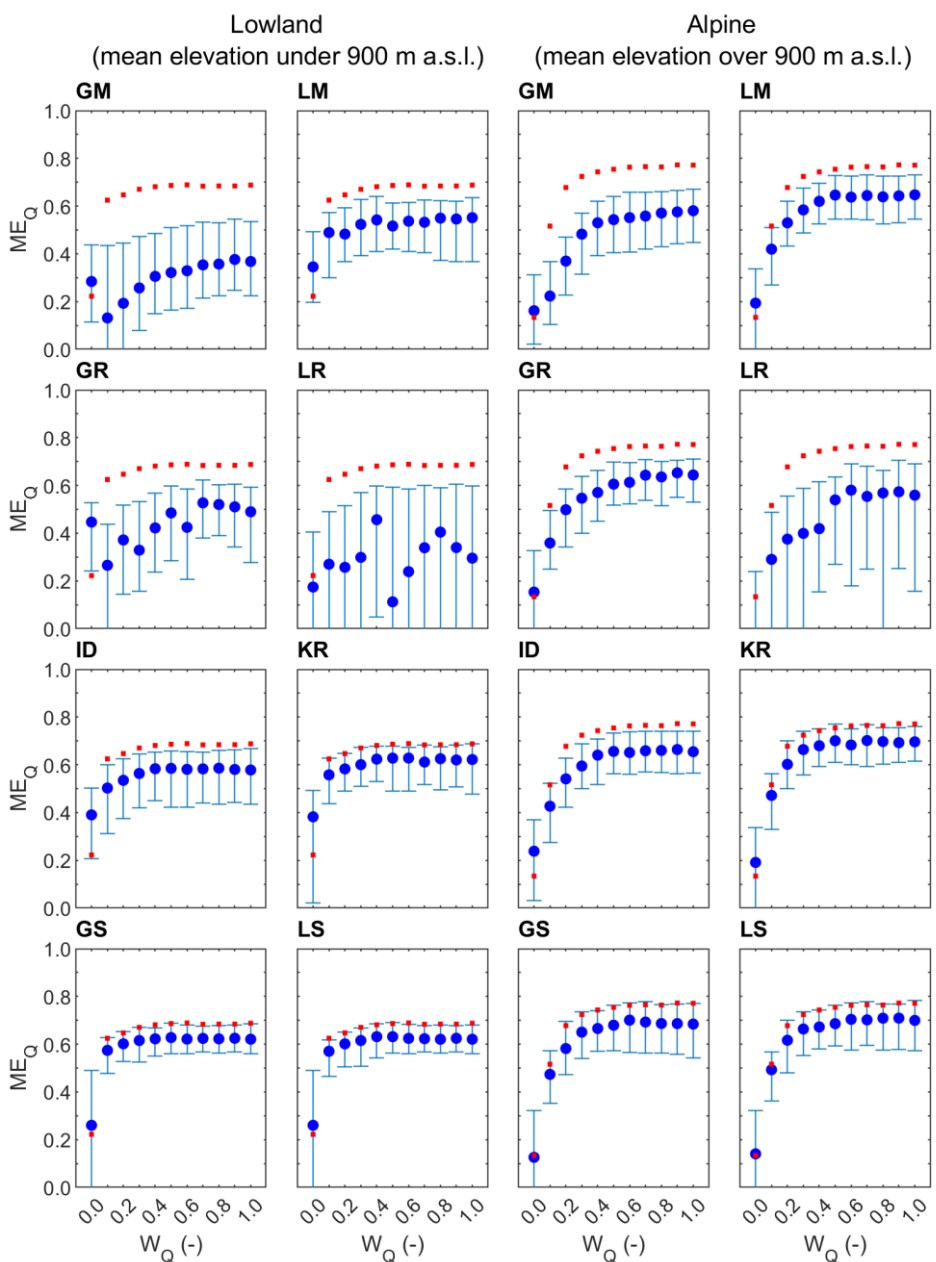

**Figure 3: Same as Fig. 2, but for the validation period (2010-2014)**

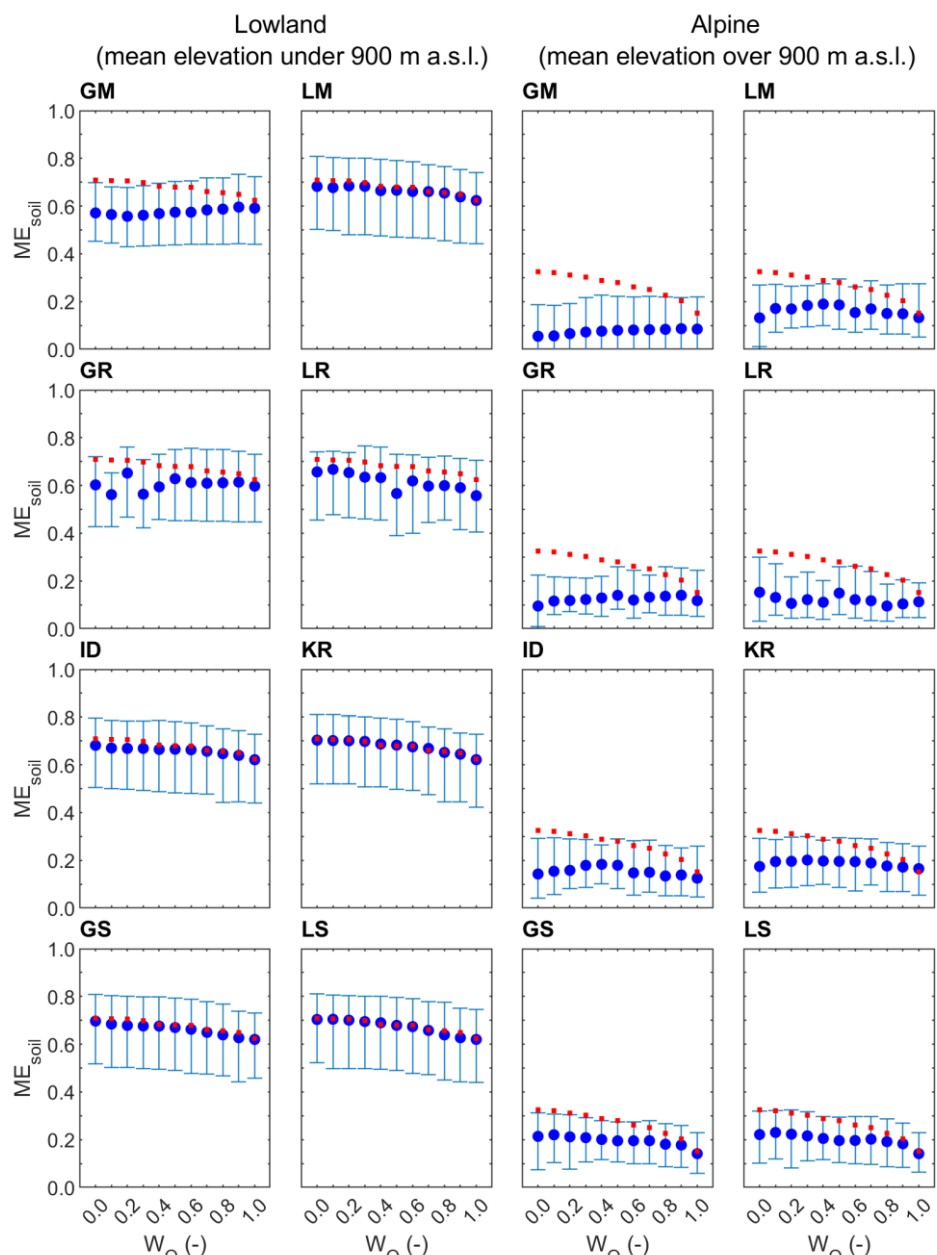

**Figure 4: Median and regional scatter of leave-one-out soil moisture correlation (Eq. 8) obtained by eight groups of parameter**
**transfer methods and eleven calibration weights for lowland (94) and alpine (119) catchments in the calibration (2007-2010, blue symbols) period. Blue circles represent the median, whiskers the 25- and 75% quantiles of leave-one-out efficiency. Red symbols show the median of at site soil moisture correlations.**

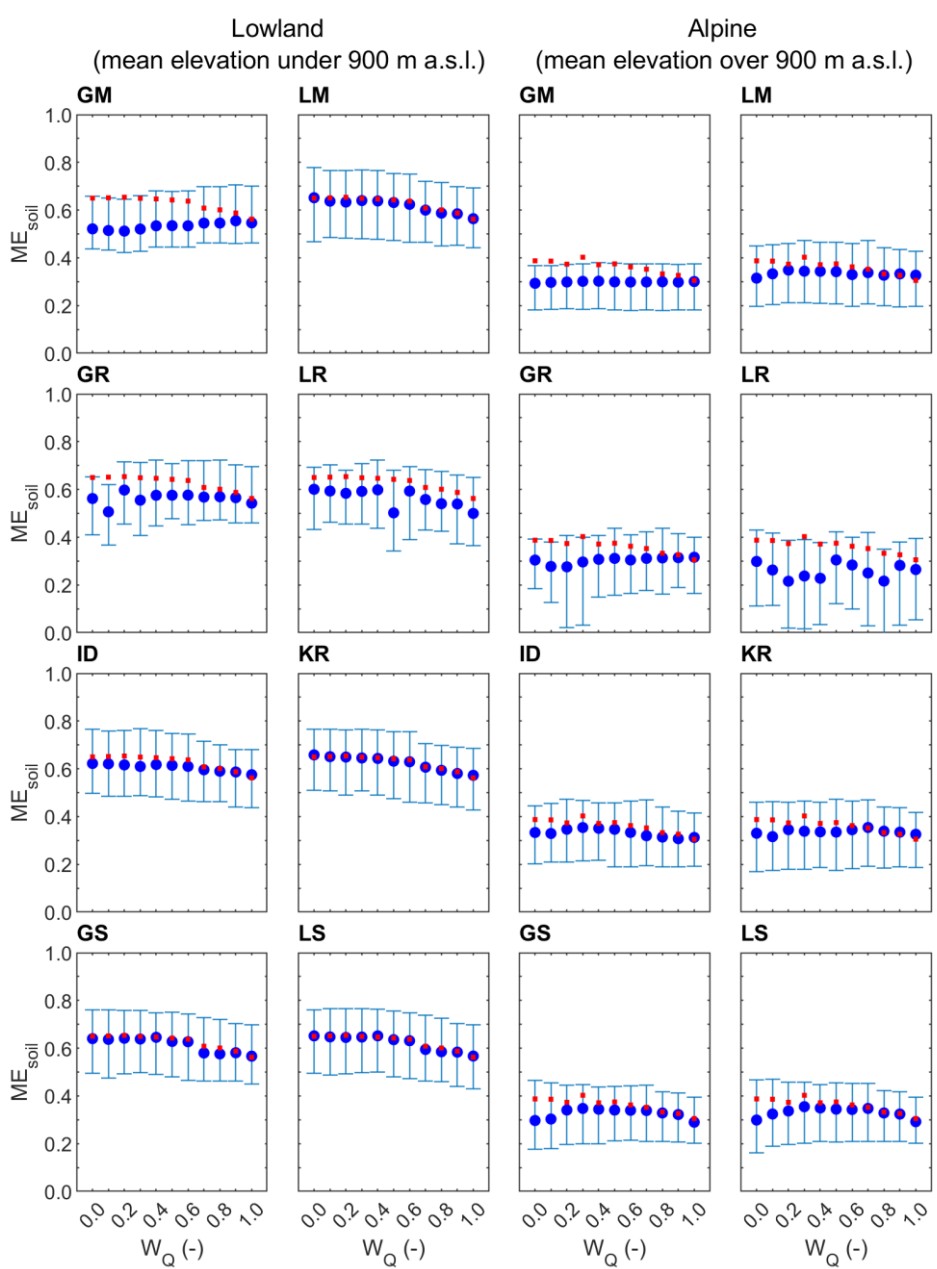

**Figure 5: Same as Fig. 4, but for the validation period (2010-2014)**

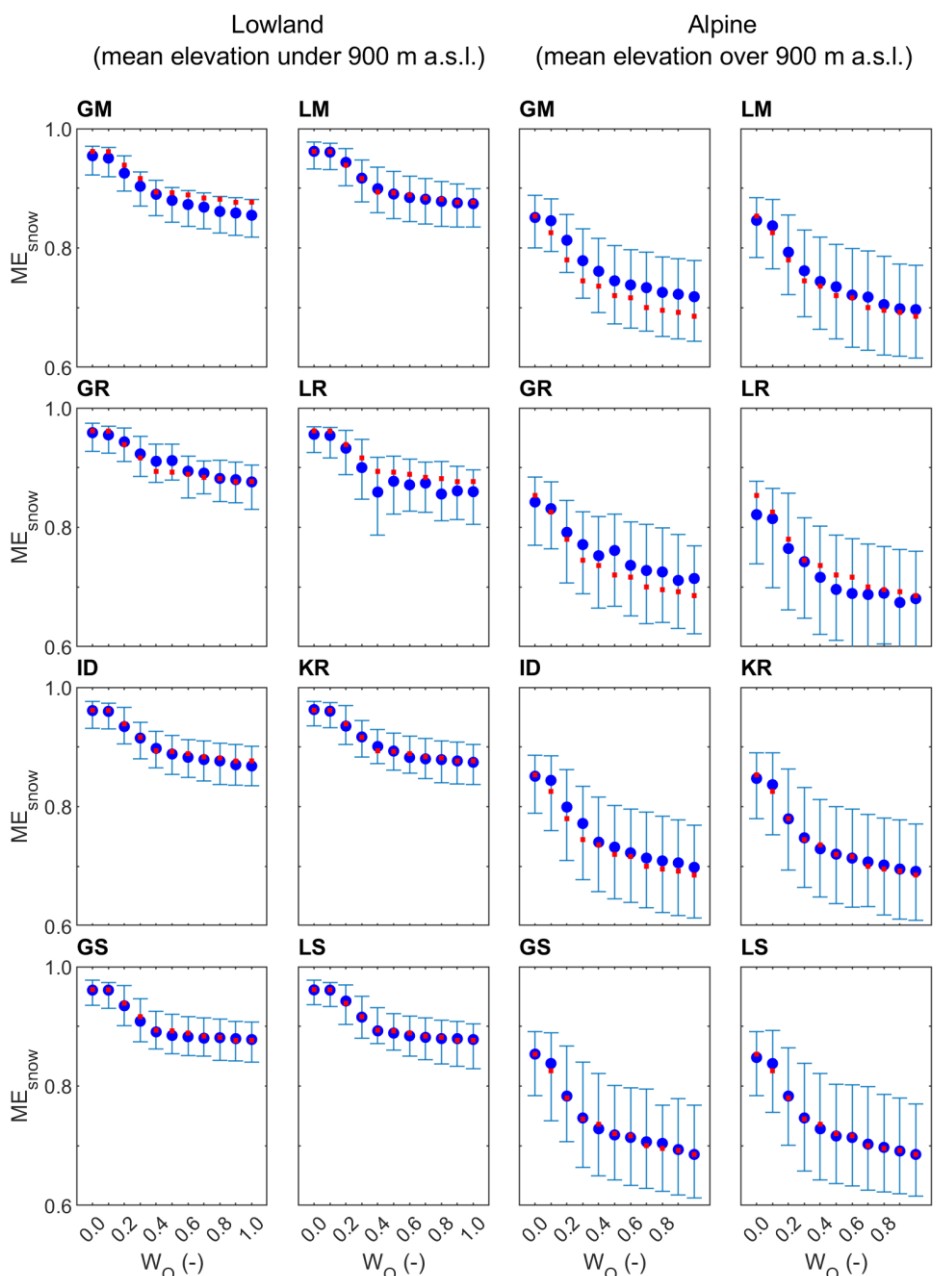

**Figure 6: Median and regional scatter of leave-one-out snow model efficiency (Eq. 9) obtained by eight groups of parameter transfer methods and eleven calibration weights for lowland (94) and alpine (119) catchments in the calibration (2000-2010, blue symbols) period. Blue circles represent the median, whiskers the 25- and 75% quantiles of leave-one out efficiency. Red symbols show the median of at site snow model efficiency.**

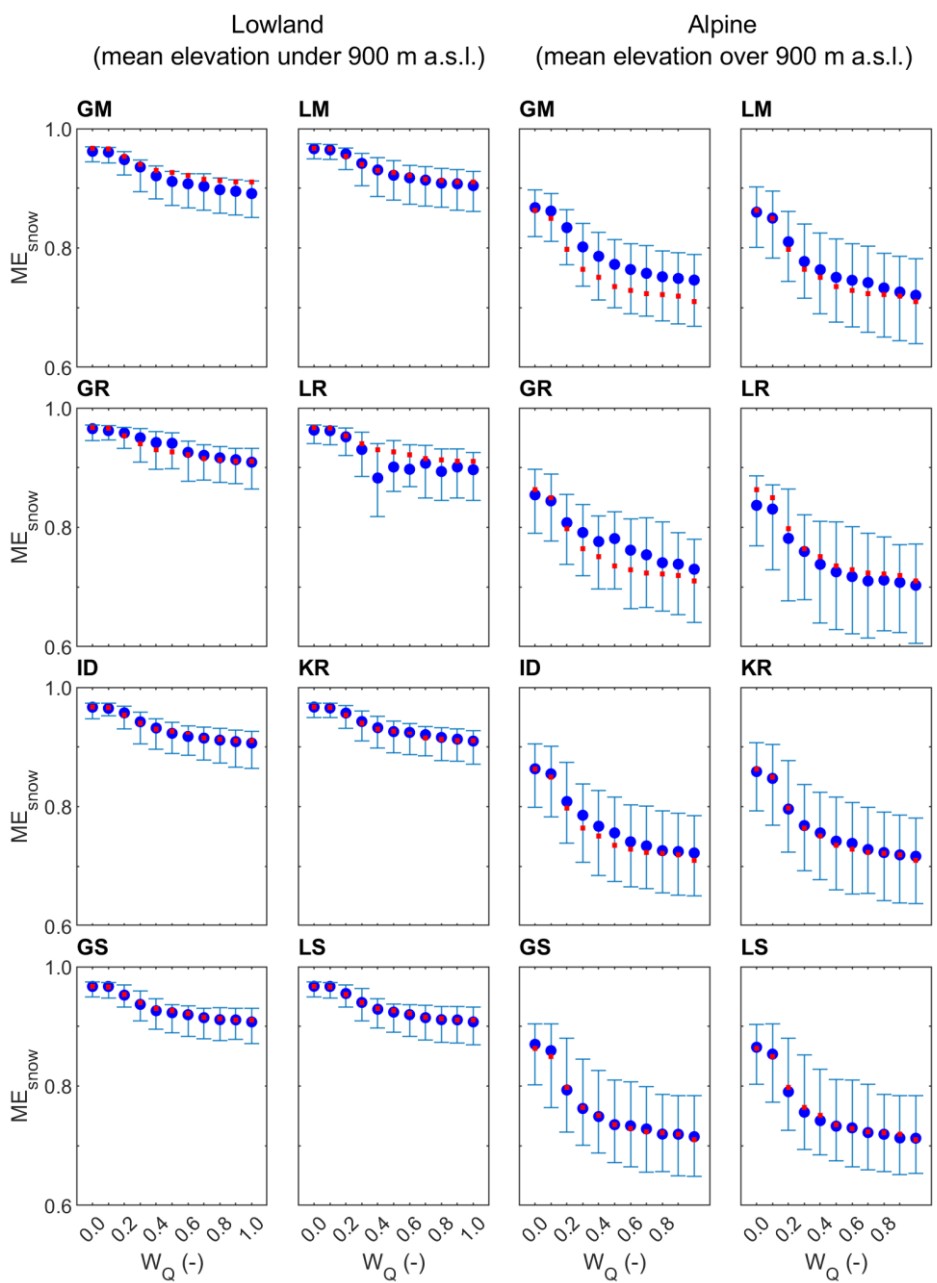

**Figure 7: Same as Fig. 6, but for the validation period (2010-2014)**

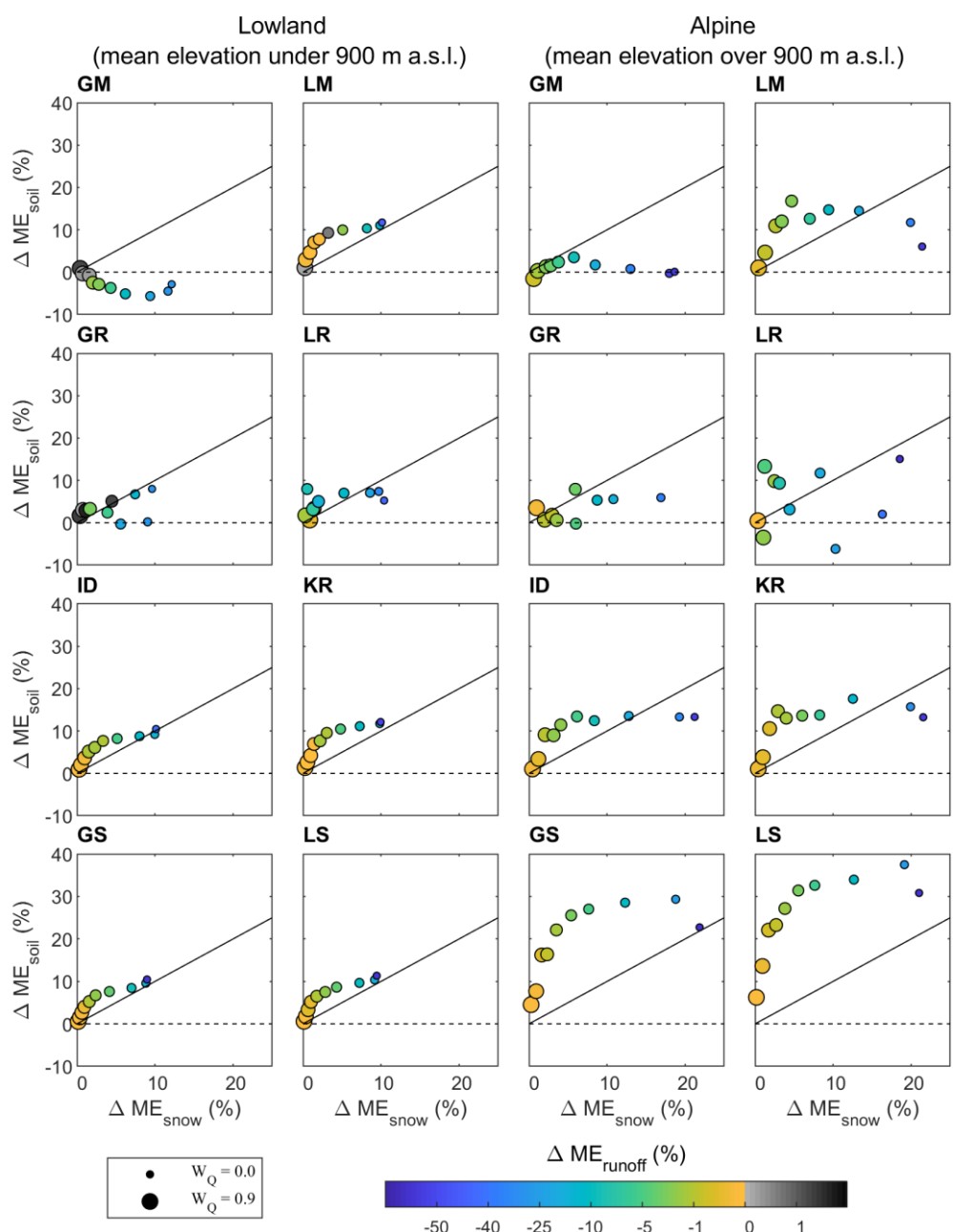

**Figure 8: Relative difference (%) in the median of snow cover (horizontal axis), soil moisture (vertical axis) and runoff (colour of symbols) efficiency between model simulations obtained by transferring model parameters calibrated by multiple-objective calibration and calibration to runoff only. The relative difference is estimated for eight model transfer methods (panels) applied in the lowland (left panels) and alpine (right panels) catchments in the calibration period 2000-2010 (soil moisture: 2007-2010).**

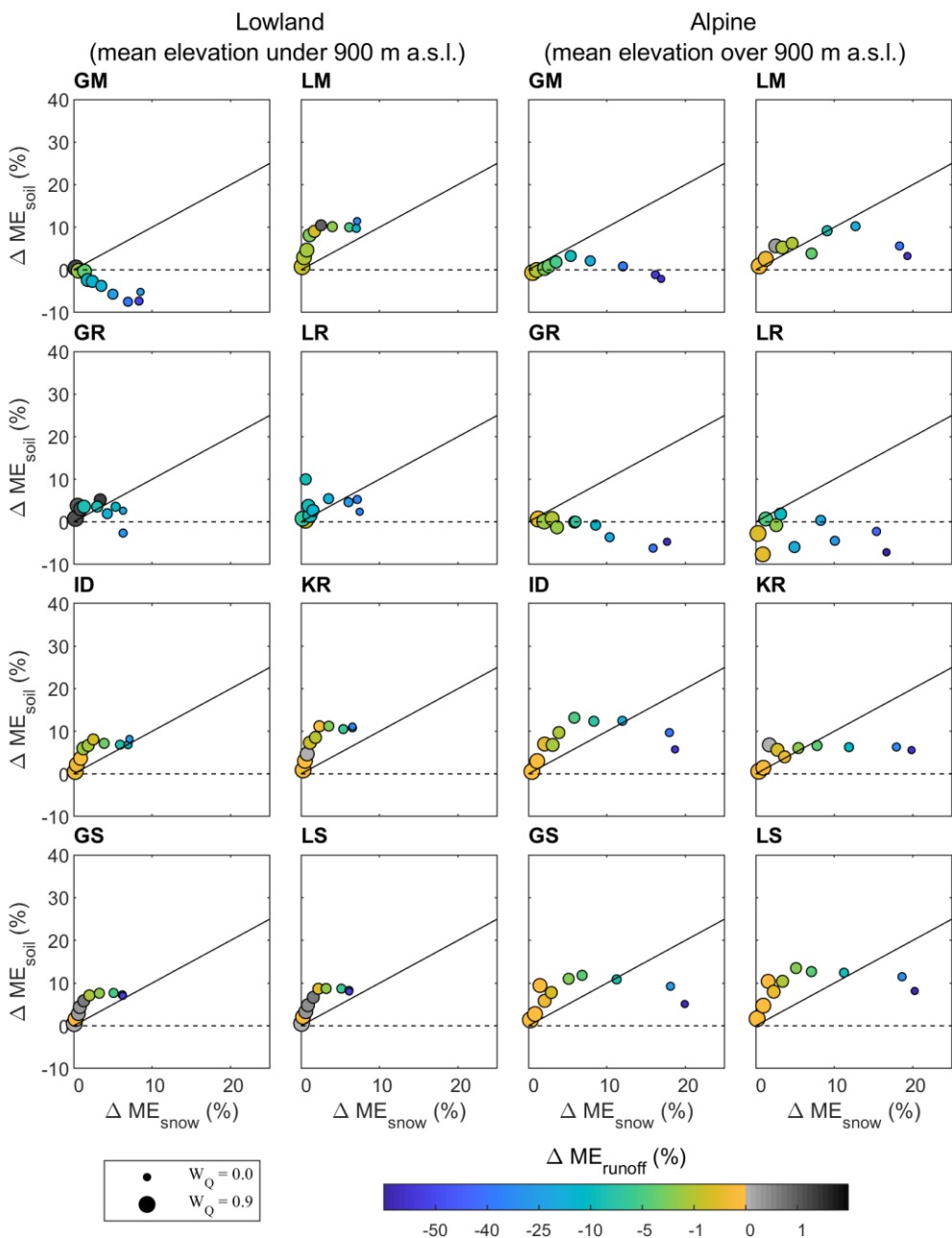

**Figure 9: Same as Fig. 8, but for the validation period (2010-2014)**
