# Peer review of "The value of satellite soil moisture and snow cover data for the transfer of hydrological model parameters to ungauged sites"

_Hydrology and Earth System Sciences, 2021_

## Referee Comment (RC1)

The study of Tong et al. focuses on testing the possibility of improving the hydrograph prediction in ungauged basins, by adding ASCAT soil moisture and MODIS snow cover data to runoff. For this aim, the study applies multi-objective calibration with changing weights between soil moisture, snow cover and runoff. Coupling the TUW model with eight typical regionalization methods, this study compares the differences and impacts of adding soil moisture and snow cover data from three aspects in 213 assumed ungauged Austria basins. The authors conclude that the calibration variant has a larger impact on runoff prediction accuracy than the selection of regionalization methods in ungauged catchments. Overall, the authors present a thorough analysis, the results seem convincing and the study is valuable for related research. However, there are several issues that still exist and need to be clarified further as indicated in the following.

First, the manuscript needs further editorial work to improve the paragraph structure and some vague expressions. The results section, figures 2-4 and tables 2-4 evaluate the prediction from two different aspects (median value and the 50% confidence interval, respectively), the text is thus suggested to set in two separate paragraphs. In addition, please pay attention to vague expressions in this manuscript, such as line 394 "This study suggests that the future evaluation of the transfer of model parameters to ungauged sites will benefit from examining what type of information will improve the calibration and transfer of model parameters related to the runoff generation and routing", which is really confusing. There are other similar sentences, so I hope the authors make a thorough change to improve the clarity of the manuscript.

Another major issue in this manuscript is that the Results section can be made more concise and to-the-point. Information presented in Figures 2-4 and table 2-4 includes both calibration and validation results, which are mostly similar, and limited text for validation result presented in current version. Thus, I would suggest that the authors focus more on one of the cases and improves the presentation of figures and tables to make sure the key messages stand out. Moving the validation information to the Supplement may be an option.

Furthermore, the conclusions are mixed with discussion in current version, which is not easy for the readers to get the key messages from the study. I would suggest that the authors conclude the findings in a separate section, and make more concise and clearer conclusions.

To conclude, I generally like the approach and methodology, but some moderate improvements are needed. I hope the authors find my comments useful and I am looking forward to an improved version of the manuscript.

Technically I have a couple of comments for current version:

(1) L138: "...with cloud cover less than a threshold 50%." Is 50% a subjective value? If so, please clarify the reason, otherwise, add the reference.

(2) L140: "... over a threshold of 25% in the zone." The same comment as above.

(3) L204-206: "...between climatic zones...the catchments were split into two groups...elevation below 900 m a.s.l. ... elevation above 900 m a.s.l.". The reference is the climatic regions, but the classification in this study is only based on elevation. Please make a clarification here, for instance, adding a table presenting the climatic statistics between two groups.

(4) L249: "Besides, to exclude invalid ASCAT measurements ... or snow cover exceeds 30 % of the pixel." Vague expression, please modify.

(5) Results: The model performance is missing. Please add a figure or table showing the assessment of model simulation accuracy in calibration and validation period. At lease, show some general

information in text.

(6) L256: "The results for the runoff weight =1.0 represent ... without using observed runoff." This information is repeated, may delete it here.

(7) L259-261: "..., for weights below 0.4. ... larger than 0.4." It is not easy to see the difference before and after 0.4 from Figure 2, please modify the text or figure.

(8) L259-261: "... In this case, ...". Here "this" is confusing, please modify the expression.

(9) L275: "The largest difference occurs ...". Please clarify "difference" here.

(10) L276: "An exception is ...". Please add reasons after this sentence.

(11) L305: "... Also here, ...". Vague expression, please modify.

(12) L320: "The results indicate the smallest difference in snow efficiency between the transferred methods". What or who is " the smallest"? please clarify and modify the expression here.

(13) L321: "A much larger difference...". Please add information about the comparison components.

(14) L323: "... between 1 and 3% ... between 8 and 17%." How did you derive this conclusion? Fig 4 shows the transfer methods individually, that readers cannot obtain this information. Please add more text information or modify the figure.

(15) L326: "... regional variability...". Please clarify its definition.

(16) L342: "... Positive efficieny values...". Please add efficiency information before this sentence, in order to connect the figure and text information.

(17) L347: "... very similar (i.e. within 1% range)...". How can the readers derive this conclusion? The legend unit in the figure is 5%.

(18) L355: "... and the improvement is larger in the alpine than the lowland catchments". In my opinion, this is an important and the most obvious finding in Figure 5 and 6, I would suggest to modify the text with an emphasis on this conclusion.

(19) L369: "... the improvement is largest..." → "... the improvement is large..."

(20) L370: "... of the efficiencies of the different..." → "... of the efficiencies between different..."

(21) L375: "... we examined all 30 transfer approaches … fewer than tested in Parajka et al. (2005)." This information is not really relevant in discussion, delete maybe.

(22) L394: This paragraph is supposed to conclusion section, the expression is not precise and clear enough for readers in current version. Please pay more attention in the logical expression and modify the conclusions more precise and clearer.

---

## Author Comment (AC2)

**Response Letter**

**The value of satellite soil moisture and snow cover data for the transfer of hydrological model parameters to ungauged sites**

Rui Tong1,2, Juraj Parajka1,2, Borbála Széles1,2, Isabella Pfeil1,3, Mariette Vreugdenhil3, Jürgen Komma2, Peter Valent2,4 and Günter Blöschl1,2

1Centre for Water Resource Systems, TU Wien, Vienna 1040, Austria

2Institute of Hydraulic Engineering and Water Resources Management, TU Wien, Vienna 1040, Austria

3Department of Geodesy and Geoinformation, TU Wien, Vienna 1040, Austria

4Department of Land and Water Resources Management, Slovak University of Technology in Bratislava, Bratislava 810 05, Slovakia

In the following document, we reproduce all the comments of the Referees in italic characters followed by our responses in blue.

**Response to referee #1**

The study of Tong et al. focuses on testing the possibility of improving the hydrograph prediction in ungauged basins, by adding ASCAT soil moisture and MODIS snow cover data to runoff. For this aim, the study applies multi-objective calibration with changing weights between soil moisture, snow cover and runoff. Coupling the TUW model with eight typical regionalization methods, this study compares the differences and impacts of adding soil moisture and snow cover data from three aspects in 213 assumed ungauged Austria basins. The authors conclude that the calibration variant has a larger impact on runoff prediction accuracy than the selection of regionalization methods in ungauged catchments. Overall, the authors present a thorough analysis, the results seem convincing and the study is valuable for related research.

We want to thank the reviewer for her/his very positive assessment of the manuscript.

However, there are several issues that still exist and need to be clarified further as indicated in the following.

First, the manuscript needs further editorial work to improve the paragraph structure and some vague expressions. The results section, figures 2-4 and tables 2-4 evaluate the prediction from two different aspects (median value and the 50% confidence interval, respectively), the text is thus suggested to set in two separate paragraphs. In addition, please pay attention to vague expressions in this manuscript, such as line 394 "This study suggests that the future evaluation of the transfer of model parameters to ungauged sites will benefit from examining what type of information will improve the calibration and transfer of model parameters related to the runoff generation and routing", which is really confusing. There are other similar sentences, so I hope the authors make a thorough change to improve the clarity of the manuscript.

Another major issue in this manuscript is that the Results section can be made more concise and to-the-point. Information presented in Figures 2-4 and table 2-4 includes both calibration and

validation results, which are mostly similar, and limited text for validation result presented in current version. Thus, I would suggest that the authors focus more on one of the cases and improves the presentation of figures and tables to make sure the key messages stand out. Moving the validation information to the Supplement may be an option.

Thanks for the suggestions. In response to this comment, we will improve the paragraph structure and revise the vague statements as suggested by the reviewer. We agree that the results for calibration and validation are similar. The reason for presenting both is to examine the split-sample performance as suggested by Klemes (1985, https://doi.org/10.1080/02626668609491024) and we will state this more explicitly in the manuscript. We will also improve the presentation of figures and tables to make sure the key messages stand out.

Furthermore, the conclusions are mixed with discussion in current version, which is not easy for the readers to get the key messages from the study. I would suggest that the authors conclude the findings in a separate section, and make more concise and clearer conclusions.

We will separate the conclusions from the discussion as suggested by the reviewer.

To conclude, I generally like the approach and methodology, but some moderate improvements are needed. I hope the authors find my comments useful and I am looking forward to an improved version of the manuscript.

We consider the comments indeed very useful. Thank you.

Technically I have a couple of comments for current version:

(1) L138: "...with cloud cover less than a threshold 50%." Is 50% a subjective value? If so, please clarify the reason, otherwise, add the reference.

This threshold was chosen on the basis of the sensitivity analysis performed by Parajka and Blöschl (2008). In response we will add "The thresholds of  $\xi_{SWE}$ ,  $\xi_C$ , and  $\xi_{SCA}$  were determined by the sensitivity analysis of Parajka and Blöschl (2008)." at the end of this paragraph.

(2) L140: "... over a threshold of 25% in the zone." The same comment as above.

Please see the response above.

(3) L204-206: "...between climatic zones...the catchments were split into two groups...elevation below 900 m a.s.l. ... elevation above 900 m a.s.l.". The reference is the climatic regions, but the classification in this study is only based on elevation. Please make a clarification here, for instance, adding a table presenting the climatic statistics between two groups.

Thanks for the suggestion, in response to this comment we have prepared a table showing the climatic statistics for the two groups which will be added to the appendix of the paper and referred to in the main text.

Table S.1. Statistics of the climatic attributes of the 94 lowland catchments and 119 alpine catchments. With abbreviation, unit, minimum, maximum, and median. The standard deviations refer to spatial variability within each catchment.

| Attribute                           | Abbrev. | Unit | Lowland (mean elevation under 900 m a.s.l.) |         |        | Alpine (mean elevation over 900 m
a.s.l.) |         |         |
|-------------------------------------|---------|------|---------------------------------------------|---------|--------|----------------------------------------------|---------|---------|
|                                     |         |      | Min.                                        | Max.    | Median | Min.                                         | Max.    | Median  |
| Mean annual precipitation           | MAP     | mm   | 728.13                                      | 1828.40 | 999.46 | 913.66                                       | 2301.84 | 1476.64 |
| Standard deviation of annual MAP    | SDAP    | mm   | 10.79                                       | 367.57  | 71.49  | 30.13                                        | 289.87  | 152.90  |
| Mean air temperature                | MAT     | °C   | 7.26                                        | 10.30   | 8.98   | -2.83                                        | 8.07    | 5.76    |
| Standard deviation of MAT           | SDAT    | °C   | 0.06                                        | 1.71    | 0.57   | 0.40                                         | 3.55    | 1.64    |
| Mean annual potential evaporation   | MEPI    | mm   | 618.36                                      | 740.45  | 690.08 | 233.49                                       | 657.01  | 563.00  |
| Standard deviation of MEPI          | SDEPI   | mm   | 4.33                                        | 77.41   | 25.25  | 21.70                                        | 162.07  | 83.33   |
| Catchment aridity index (MEPI/MAP)  | CAI     | -    | 0.36                                        | 0.98    | 0.66   | 0.18                                         | 0.69    | 0.37    |
| Standard deviation of aridity index | SDAI    | -    | 0.01                                        | 0.31    | 0.06   | 0.02                                         | 0.18    | 0.09    |

(4) L249: "Besides, to exclude invalid ASCAT measurements ... or snow cover exceeds 30 % of the pixel." Vague expression, please modify.

We modified the sentence as "Besides, to exclude invalid ASCAT measurements affected by snow and frozen ground, soil moisture is masked as no data when soil temperatures at a soil depth of 0-7 cm are below 1°C or snow cover exceeds 30 % of the pixel with the information from the ECMWF Copernicus Climate Service (C3S) ERA5-Land."

(5) Results: The model performance is missing. Please add a figure or table showing the assessment of model simulation accuracy in calibration and validation period. At lease, show some general information in text.

In response to this comment, we will indicate the at site calibration performance in the figures and text, as suggested by the reviewer. We will also add an evaluation of the loss of performance of the different regionalization methods.

(6) L256: "The results for the runoff weight =1.0 represent ... without using observed runoff." This information is repeated, may delete it here.

Thank you, we will remove the repetition.

(7) L259-261: "..., for weights below 0.4. ... larger than 0.4." It is not easy to see the difference before and after 0.4 from Figure 2, please modify the text or figure.

In response to this comment, we will modify Figure 2 and add X axis grid lines. We believe this will indicate the position of the performance for the selected weights more clearly.

(8) L259-261: "... In this case, ...". Here "this" is confusing, please modify the expression.

We will modify the sentence as follows: "For wQ larger than 0.4 the differences between the transfer methods are larger, ..."

(9) L275: "The largest difference occurs ...". Please clarify "difference" here.

We will add "between the local and global methods" after "difference".

(10) L276: "An exception is ...". Please add reasons after this sentence.

In response to this comment, we will add the following explanation: "An exception is the regression of model parameters, which has a larger runoff efficiency for the global than the local approach. The reason is a larger correlation between model parameters and catchment attributes estimated from all catchments. For example, for wQ=0.4, the median of the correlation between model parameters and catchment attributes for the local regression varies between 0.22 and 0.65. For the global regression approach, the median is larger and varies between 0.70 and 0.88."

(11) L305: "... Also here, ...". Vague expression, please modify.

We will modify the sentence as follows: "The best transfer methods in alpine catchments are local and global similarity and kriging. The median correlation between modelled and satellite soil moisture is however small and varies between 0.14 and 0.22."

(12) L320: "The results indicate the smallest difference in snow efficiency between the transferred methods". What or who is " the smallest"? please clarify and modify the expression here.

We will modify the sentence as follows: "The results indicate that the variability and differences between the regionalization approaches are the smallest for snow efficiency."

(13) L321: "A much larger difference...". Please add information about the comparison components.

We will modify the sentence as follows "A much larger difference and impact on snow efficiency has the runoff weight used in model calibration."

(14) L323: "... between 1 and 3% ... between 8 and 17%." How did you derive this conclusion? Fig 4 shows the transfer methods individually, that readers cannot obtain this information. Please add more text information or modify the figure.

We compared how the snow efficiency varies between regionalization methods and/or between the different runoff weights used in model calibration. The difference/variability in the median snow cover efficiency obtained by different regionalization methods, but for the same runoff weight (used in model calibration) is smaller than the difference (variability) in median snow model efficiency obtained by individual regionalization method across different runoff weights. For example the median of snow efficiency for the runoff weight 0.4 varies between eight groups of regionalization methods between 0.72 (local regression) and 0.74 (global mean) in the alpine and between 0.88 (local regression) and 0.91 (global regression) in the lowland catchments. This variability in medians is about 3%. The variability in medians for one regionalization approach (e.g. kriging) is between 0.87 (runoff weight 1.0) and 0.96 (runoff weight 0.0), which is in relative terms approximately 9% variability. In order to allow a more direct comparison of the efficiency values for the same runoff weight or differences between runoff weights we will modify the Figure 4 by adding gridlines, showing more precisely the efficiency values for different runoff weights. (15) L326: "... regional variability...". Please clarify its definition.

Regional variability refers to differences in model efficiency between the catchments. We will introduce this term in section 4.

(16) L342: "... Positive efficient values...". Please add efficiency information before this sentence, in order to connect the figure and text information.

We will rephrase the paragraph as follows: "We compared the efficiencies of the predictions obtained by transferring model parameters from multiple-objective calibration (i.e. wQ

---

## Author Response (AR1)

**Response Letter**

**The value of satellite soil moisture and snow cover data for the transfer of hydrological model parameters to ungauged sites**

Rui Tong[1,2], Juraj Parajka[1,2], Borbála Széles[1,2], Isabella Pfeil[1,3], Mariette Vreugdenhil[3], Jürgen Komma[2], Peter Valent[2,4] and Günter Blöschl[1,2]

[1]Centre for Water Resource Systems, TU Wien, Vienna 1040, Austria
[2]Institute of Hydraulic Engineering and Water Resources Management, TU Wien, Vienna 1040, Austria
[3]Department of Geodesy and Geoinformation, TU Wien, Vienna 1040, Austria
[4]Department of Land and Water Resources Management, Slovak University of Technology in Bratislava, Bratislava 810 05, Slovakia

In the following document, we reproduce all the comments of the Referees in italic characters followed by our responses in blue.

**Response to editor**

*Thank you for your responses to the reviewer comments. From what I can understand you aim to make some useful changes that will improve the manuscript. I would say the reviewers are mixed in their assessment of your paper but I believe with major changes to your approaches and results then the paper could become a useful contribution. I do accept that applying spatial data and in a multi-objective approach to the regionalization problem warrants a novel enough approach to be included in the literature,*

We would like to thank the Editor for the evaluation of the manuscript.

*however I do wish to make the following points for the next evaluation of this paper:*
*1) I do not agree that in your response to referee #2 that you are really dealing with the core uncertainties in this process. I'd like to see some significant justification as to why you can possibly consider that the adjustment of weights is fundamentally the most important source of uncertainty that your experimental design faces or really that this in some way relates to the core matters the reviewer is trying to get you to address. You are using multi-objectives here, and spatial information, you are currently treating them all as if they are deterministic. I think when we use multi response data we have to care and mind what they represent and their accuracy. Here you provide no evidence to justify your methods and this needs to change*

We agree with the Editor about the importance of providing insights into the various uncertainty sources of the data and the experimental design. During the conceptualization of the analysis we considered the following potential sources of uncertainty:

  (a) model inputs
  (b) model structure
  (c) accuracy of satellite data

(d) model calibration
(e) model parameter regionalization

[revised manuscript text omitted]

*2) Secondly and partly related to the above and as I have noted in my editorial review, the methods section is extremely poor (still) on explaining how you are comparing these spatial information to your model framework and how commensurate they are (and the issues and assumptions that have to be dealt with). If your paper is a valuable contribution to introducing this type of spatial information into the regionalization process then I expect the paper to give this full and detailed consideration of the steps needed to make those comparisons effective and 'plausible' Here there is a smoke screen of how lower resolution information is disaggregated and related to a model that has only spatial elevation bands and homogeneous parameters for each catchment. The paper does not attempt to explain in detail the approach used to compare these quantities nor explains how soil moisture (for a certain depth average) can be related to a potential different depth average of a model conceptualization. I don't mind if this is fully detailed in appendices etc. but this has to be massively improved with appropriate figures and explanations. In conjunction with this there is almost no evaluation as to the trade offs and parameterizations across catchments to how well the model does compared to this information. This again needs to improve as the plots currently are too summarized to explain the real value of the information in the multi-objective analyses and thus the value to the regionalization approach.*

In response to this comment, we have extended the Methods section and added a Supplement section as suggested by the Editor. This revision provides more detailed information on why and how we have estimated the agreement between satellite and modelled soil moisture and a detailed description of how we relate modelled and satellite soil moisture.

Methods section:

"The rationale behind selecting the Pearson correlation as a measure of agreement is that it assesses the spatial and temporal correspondence of the satellite soil moisture and simulated root zone soil moisture time series. At the spatial resolution of original ASCAT dataset (ca. 12.5 km), the satellite estimates of root zone soil moisture reflect mainly regional rainfall and melt processes patterns, and are thus more closely related to altitudinal zonality than to morphometric

characteristics of the terrain that operate at smaller scales. The calculation of $O_{SM}$ from soil moisture averages for elevation zones thus allows representing the agreement in regional and seasonal soil moisture patterns. Choice of a correlation coefficient has the advantage of not being sensitive to the units. In a preliminary analysis, we tested different methods for calculating $O_{SM}$ and found that the $O_{SM}$ combining soil moisture estimated from different elevation zones better describes the soil moisture agreement than the correlation between soil moisture estimates averaged at the catchments scale (see Supplement, Fig. S3). Particularly in the alpine regions, correlation calculated from catchment averages masks the spatial variability in the agreement between ASCAT and hydrologic root zone soil moisture estimates. A similar approach has been used in previous studies (e.g., Parajka et al., 2006; Gruber et al., 2020; Beck et al., 2021). A more detailed description of the calculation of soil moisture agreement is presented in the Supplement."

Supplement section:

Soil moisture is one of the key controls of runoff response. Past studies have used ground soil moisture measurements to provide insight into spatial and temporal soil moisture patterns and their relation to terrain, and soil and vegetation characteristics (e.g. Bardossy and Lehmann, 1998, Western and Blöschl, 1999). However, ground-based measurements have spatial supports of only a few centimetres, and logistically, they can only cover relatively small areas. This makes it very difficult to estimate meaningful spatial averages over medium-sized to large catchments. Alternative more relevant for larger catchments are hydrological models and satellite observations (Babaeian et al., 2019). The main advantage of using hydrological models is that they explicitly represent areal averages, and soil moisture simulated by these models is considered vertically representative over the entire root zone (i.e. the critical zone for runoff generation) but they always need calibration for accurately representing hydrological processes in a particular case (Blöschl and Grayson, 2002).

The TUWmodel used in this study is a conceptual hydrologic model, which simulates soil moisture in the root zone. The changes in the soil moisture state result from changes in snowmelt, rainfall, evapotranspiration and runoff generation contributions. The parameterization of soil moisture and runoff generation has three model parameters ($FC$, $Beta$, $LP$), which are calibrated. The relationship between rainfall, melt, soil moisture storage and runoff generation is described by a non-linear function, which is an empirical curve that connects effective precipitation to simulated soil moisture storage and the model parameter field capacity ($FC$) (Bergström and Lindström, 2015). The contribution of rain ($P_R$) and snowmelt ($M$) to runoff is calculated by an explicit scheme as a function of the soil moisture $SSM$ in the root zone, using the following non-linear relationship:

$$\Delta S_{UZ} = \left(\frac{S_{SM}}{FC}\right)^{Beta} \cdot (P_R + M),$$

where $FC$ is the maximum soil moisture storage and $Beta$ is a parameter that controls the characteristics of runoff generation. Similar concepts can, for example, be found in the Xinanjiang model (Zhao, 1992) and the VIC model (Liang and Lettenmaier, 1994). For a full

description of the TUWmodel and its implementation see Viglione and Parajka (2020), Astagneau et al. (2021) and Jansen et al. (2021).

Satellite observations similarly provide an integral value over an area which allows direct comparisons with hydrologic models. Most satellite datasets are available globally with relatively high temporal resolution, so they are also suited for ungauged catchment predictions. However, microwave-based datasets have limited penetration depths and poor estimation under dense vegetation, on frozen ground and for snow-covered conditions. Because of the limited penetration depth of a few centimetres, further processing is needed to obtain soil moisture estimates over a deeper soil layer.

The satellite estimates of root zone soil moisture used in this study are based on the change detection method of Wagner et al. (1999) which relates surface soil moisture and satellite backscatter. The surface soil moisture is determined by extrapolating the backscatter coefficient to a reference angle of 40° and accounting for surface roughness and vegetation characteristics. A simple two-layer water balance model then estimates the root zone soil moisture. The first layer represents the remotely sensed topsoil layer, and the second layer represents a reservoir connected to the surface layer. It is assumed that the surface wetness observations from the scatterometer reflect the high soil moisture dynamics due to precipitation, evaporation, and surface runoff and indicate the wetting and drying trend of the moisture content in the lower soil profile. The water flux between the two layers is assumed to be proportional to the volumetric water content in the surface layer and the reservoir. The result of this model is a Soil water index, which represents the profile soil moisture in relative units ranging between wilting point and field capacity. This method has been validated and compared with ground-based and modelled root-zone soil moisture estimates in numerous studies (e.g. Paulik et al., 2014). It has become a part of the processing algorithms providing operational and experimental soil moisture products, such as S1ASCAT used in this study (Bauer-Marschallinger et al., 2018).

One of the aims of this study is to compare the hydrologic model and satellite soil moisture predictions in ungauged basins. The procedure consists of transferring model parameters to ungauged basins, running the model, and estimating runoff, soil moisture and snow cover. We use a semi-distributed hydrologic model for the modelling and calculate the soil moisture and snow cover in individual elevation zones in each catchment. The catchments are partitioned into elevation zones of 200 m vertical width. The main idea of our approach is to keep the number of model parameters small (to allow an effective transfer to ungauged sites), but to represent the spatial (mostly altitudinal) variability of runoff processes, including snowmelt in alpine areas. Our approach uses lumped model parameters (i.e. the same parameters in all elevation zones of a catchment), but the model inputs and state variables differ between elevation zones. This methodology has been widely used in the past (e.g. Paris Anguela et al., 2008, Parajka et al., 2009).

The individual steps of the methodology are documented in Figs S1-S4. Fig. S1 shows an example of the regional patterns of root zone soil moisture estimated from the ASCAT satellite, indicating that the spatial resolution reflects mainly the large scale rainfall patterns and antecedent melt processes, rather than the morphometric characteristics of the terrain (e.g. differences between concave and convex landforms).

[Figure]

Figure S1. Relative root zone soil moisture from ASCAT on May 15, 2016 in Austria. Grey colour indicates masking because of snow cover.

Figure S1 shows higher soil moisture in central and western (alpine) parts of Austria due to rainfall on May 14, 2016 and preceding snowmelt than in the eastern lowlands. Wetter soils in the South-east reflect local rainfall events on May 14 and 15. Both the seasonal precipitation and melt processes have strong altitudinal variability, so we decided to estimate the agreement in soil moisture for individual elevation zones in each catchment. We extracted for each day the average satellite soil moisture in each elevation zone in each catchment. The example in Figure S2 shows the S1ASCAT root-zone soil moisture averages for different elevation zones at the top and observed daily discharge at the bottom.

[Figure]

Figure S2. Time-series of observed S1ASCAT soil moisture in different elevation zones and observed discharge for the Pramerdorf-Pram catchment (341 km²) in Upper Austria.

In a next step we considered (in turn, leave one out) each catchment as ungauged and transferred calibrated model parameters to it by using different regionalization methods. The model parameters had been calibrated in a previous study of Tong et al. (2021) using a multiple-objective framework. We tested 11 different sets of model calibrations representing different runoff weightings and satellite snow cover and soil moisture objective functions. While the weight $w_Q$=1 represents a traditional calibration to runoff only, $w_Q$=0 represents a calibration to snow cover and soil moisture only. Values of $w_Q$ between 0 and 1 represent different tradeoffs between these objectives. Regionalization model performance in each catchment was then

evaluated against runoff and satellite data. The soil moisture efficiency compares the correlation (OSM) between the simulated relative root zone soil moisture and the ASCAT snow water index. In a preliminary testing phase, we tested different methods for calculating the OSM agreement. We found that the OSM combining soil moisture estimated from different elevation zones allows more robust description of the OSM agreement than the correlation between soil moisture estimates averaged over the entire catchment (Fig. S3).

[Figure]

Figure S3. Comparison of the Pearson correlation coefficient (MEsoil) estimated from the mean catchment averages (red symbols) and from elevation zone averages (blue symbols) in 213 catchments. The black bars show the variability of Pearson correlation calculated for the elevation zones within each catchment. The Pearson correlation is estimated between ASCAT root zone soil moisture and hydrologic model relative root zone soil moisture in the calibration period.

Particularly in the alpine (higher altitude) regions, correlations calculated from the catchment averages of soil moisture estimates hide the spatial variability in the agreement between ASCAT and hydrologic root-zone soil moisture estimates, because in catchments with large altitudinal variability, the correlation between catchment averages is often large (red symbols in Fig. S3), but the soil moisture agreement in higher elevation zones (lines representing the range of correlation in Fig. S3) is much smaller. We thus decided to estimate the correlation for elevation zones rather than catchment averages. The final correlation coefficient is calculated as average over all elevation zones for every day where soil moisture is available. Days for which elevation zone average satellite soil moisture cannot be estimated (due to missing pixel values that indicate

snow cover or frozen ground) are excluded from the correlation estimation. An example of the soil moisture agreement (correlation) for the Pramersdorf-Pram catchment in the calibration and validation period is presented in Figure S4.

[Figure]

Figure S4 Example of soil moisture agreement (correlation) for the Pramersdorf-Pram catchment in the calibration and validation periods.

---

## Author Response (AR2)

**Response Letter**

**The value of satellite soil moisture and snow cover data for the transfer of hydrological model parameters to ungauged sites**

Rui Tong[1,2], Juraj Parajka[1,2], Borbála Széles[1,2], Isabella Greimeister-Pfeil[1,3], Mariette Vreugdenhil[3], Jürgen Komma[2], Peter Valent[2,4] and Günter Blöschl[1,2]

[1]Centre for Water Resource Systems, TU Wien, Vienna 1040, Austria
[2]Institute of Hydraulic Engineering and Water Resources Management, TU Wien, Vienna 1040, Austria
[3]Department of Geodesy and Geoinformation, TU Wien, Vienna 1040, Austria
[4]Department of Land and Water Resources Management, Slovak University of Technology in Bratislava, Bratislava 810 05, Slovakia

In the following document, we reproduce all the comments of the Referees in italic characters followed by our responses in blue.

**Response to Referee #1**

*I did not find the corresponding modification to my previous comment (5). In my opinion, the model performance assessment in calibration and validation period is the base information for deep or further assessment. I would like to suggest the authors to present this information in the final version of the manuscript.*

**In response to this comment and the comment of the second reviewer, we have modified Figs. 2-4. We split the presentation of calibration and validation efficiencies into separate figures, plot the variability in the form of bars and, as suggested by the reviewer, we plot also the median of at site model efficiency for comparison.**

**Response to Referee #2**

*In this manuscript the authors analyze the potential value of constraining models with observations of multiple variables, i.e. stream flow, snow cover and soil moisture, for the spatial model transferability.*

*This is a highly relevant topic that has so far remained under-explored in literature. The experiment is well designed, follows a logical sequence and a sound, exhaustive methodological approach, thereby providing a comprehensive picture of the effect of the tested multi-variable calibration strategy on model transferability – i.e. >200 catchments, 11 different calibration weights for 8 regionalization techniques, further discretized into low- and upland regions as well as into model calibration and evaluation periods. The results are clearly described and documented. Altogether the analysis is impressively complete.*

*However, the authors could further strengthen their manuscript by more clearly emphasizing their overall objective in the description and discussion of the results but also in the Figures. In*

*its current state, the text and figures may give the reader the impression that the emphasis and novelty is on the 8 transfer methods, as also reflected in the reaction of one of the previous reviewers. I believe that this was not the intention of the authors and I think the manuscript would strongly benefit from re-directing the attention of the reader to the actual objective: the effect of multi-variable calibration on model transferability. To achieve this, I am convinced that this requires only some minor twists and rephrasing in how results are presented and discussed in the text/figures.*

**We would like to thank Prof. Hrachowitz for his positive and constructive evaluation of our paper. In response to this comment, we have tried to improve the presentation of the results.**

*Minor comments:*

*Tables 3, 4 and 5: I find these tables not really intuitive. Why is this information not added in Figures 2, 3 and 4, e.g. as error bars? This would make it much more convenient to appreciate this information.*

**Thanks. In response to this comment, we have revised Figs. 2-4 as suggested by the reviewer. To keep the clarity of presentation, we split the results of calibration and validation efficiencies. We present the variability of the efficiency in the form of bars and (as suggested by another reviewer) we plot in the figures also the median of at site calibration and validation efficiency.**

*Figures 5 and 6: the irregular steps in the color-scale are a bit confusing. It may be helpful for the reader to use a continuous color-scale.*

**In response to this comment, we have changed the colour scale (as suggested).**

*l.406: perhaps worth mentioning other studies that report similar conclusions. Here just a few examples from our group, but there quite some from other groups as well: Dembele et al., 2020; Hulsman et al., 2021 (please only see those as mere suggestions and do not feel obliged to cite them)*

**Thanks for recommendation and reference to a recent interesting study. One of the suggested references is already cited, the second was added as suggested.**